# Convergence Aspects of Hybrid Kernel SVGD

**Anson MacDonald**                                                    *anson.macdonald@unsw.edu.au*
*School of Mathematics and Statistics*
*University of New South Wales*

**Scott A. Sisson**                                                    *scott.sisson@unsw.edu.au*
*School of Mathematics and Statistics*
*University of New South Wales*

**Sahani Pathiraja**                                                    *s.pathiraja@unsw.edu.au*
*School of Mathematics and Statistics*
*University of New South Wales*

**Reviewed on OpenReview:** *https://openreview.net/forum?id=JZkbMSQDmD*

## Abstract

Stein variational gradient descent (SVGD) is a particle-based approximate inference algorithm. Many variants of SVGD have been proposed in recent years, including the hybrid kernel variant (h-SVGD), which has demonstrated promising results on image classification with deep neural network ensembles. By framing h-SVGD as a kernelised Wasserstein gradient flow on a functional that is not the Kullback-Leibler divergence, we demonstrate that h-SVGD does not converge to the target distribution in the mean field limit. Despite this theoretical result, we provide intuition and experimental support for the ability of h-SVGD to improve variance estimation in high dimensions. Unlike other SVGD variants that also alleviate variance collapse, this is achieved at no additional computational cost and without further assumptions on the posterior.

## 1 Introduction

Stein variational gradient descent (SVGD) is a variational inference algorithm that generates samples from a target probability density (Liu & Wang, 2016). It has proven useful in many tasks in Bayesian inference and machine learning. SVGD evolves an interacting particle system until the particles resemble a sample from a target density. The dynamics of this system include a driving term that moves particles to regions of high probability, and a repulsive term that repels particles from one another. This repulsive term prevents particles from converging to the same mode. It has been shown that within a unit ball of a reproducing kernel Hilbert space (RKHS), the SVGD update direction optimally reduces the Kullback-Leibler (KL) divergence between the target density and the approximating density (Liu & Wang, 2016). The reproducing kernel of this RKHS appears in both the driving and repulsive terms, making the choice of kernel a key ingredient for SVGD.

The theoretical properties of vanilla SVGD have been studied extensively. Liu showed that the empirical measure of the particles converges weakly to the target distribution (Liu, 2017). SVGD in the mean field regime has been described as a gradient flow on the KL divergence (Liu & Wang, 2016) and the chi-squared divergence (Chewi et al., 2020). Furthering this geometric point of view, Duncan et al. (2023) developed the Stein geometry along with its associated tangent spaces and geodesics, leading to guidelines for choosing kernels to improve convergence. The existence and uniqueness of the solution to the Stein partial differential equation (PDE) has been established (Lu et al., 2019) along with various descent lemmas bounding the decrease in KL divergence at each iteration (Liu, 2017; Korba et al., 2020; Salim et al., 2022).

SVGD is known to suffer from the curse of dimensionality through variance collapse (Wang et al., 2018; Ba et al., 2019) whereby the marginal variances of the particles underestimate the true marginal variances of the target in high dimensions. Zhuo et al. (2018) explained that this phenomenon is due to the size of the repulsive term of the update direction scaling inversely with dimension. This enables the driving term to dominate in high dimensions, thereby forcing particles to converge to the mode(s) of the target. This insight suggests that strengthening the repulsive term in SVGD should lead to better variance estimation, an idea which we explore in Sections 3 and 4.

Many variants of SVGD have also been proposed in recent years, some offering improvements and others providing generalisations. Riemannian SVGD (Liu & Zhu, 2018) generalises SVGD by allowing for target densities on Riemannian manifolds, not just Euclidean spaces. Matrix SVGD (Wang et al., 2019) replaces the scalar valued kernel with a matrix valued kernel to incorporate preconditioning information and speed up particle exploration. Message passing SVGD (Zhuo et al., 2018) and graphical SVGD (Wang et al., 2018) focus on target densities that factorise according to a graph structure. This approach reduces variance collapse in high dimensions by converting the problem to a collection of low dimensional problems. Projected SVGD (Chen & Ghattas, 2020), sliced SVGD (Gong et al., 2021) and Grassman SVGD (Liu et al., 2022) also mitigate the issue of variance collapse by updating particles within lower dimensional subspaces, which comes at the expense of additional computation.

In this work, we study a variant called hybrid kernel Stein variational gradient descent (h-SVGD). The name comes from its use of two distinct kernels for the driving and repulsive terms with the aim of mitigating variance collapse. This variant was originally proposed by D'Angelo et al. (2021) in the context of training deep neural network ensembles by sampling from the distribution of network parameters. In that setting, two particles may parameterise networks with very similar outputs despite being far apart in the weight space. Their insight was to encourage functional diversity between networks in the ensemble by using a standard kernel in the driving term, but a functional kernel in the repulsive term. In this neural network ensemble setting, h-SVGD demonstrated better performance than other variants on image classification. Annealed SVGD (D'Angelo & Fortuin, 2021) may also be considered an example of h-SVGD. In this variant, the driving kernel is a scalar multiple $\gamma(\ell) \in [0, 1]$ of the repulsive kernel, and this factor $\gamma(\ell)$ gradually increases to 1 as the iteration $\ell$ increases. Numerical experiments show that annealed SVGD improves the ability of particles to escape local modes. Scaling one of the update terms has also been used as a computational technique to aid other SVGD variants when training Bayesian neural networks (Gong et al., 2021).

Although preliminary numerical experiments have shown the benefits of h-SVGD (D'Angelo et al., 2021), the theoretical results for SVGD do not directly apply in the hybrid kernel setting due to the presence of a second kernel. In this paper, we address this theoretical gap and reinforce the practical benefits of h-SVGD through the following contributions.

- We establish existence of a solution to the hybrid Stein PDE and a kernelised Wasserstein gradient flow interpretation. Through the study of this gradient flow, we demonstrate that h-SVGD does *not* converge to the target distribution in the mean field limit.

- We also quantify the rate of dissipation of the gradient flow functional and develop a discrete time version of this result, otherwise known as a descent lemma.

- Despite not converging to the target distribution, we demonstrate through numerical experiments that h-SVGD can mitigate variance collapse in the finite particle regime at negligible additional cost, whilst remaining competitive at high dimensional inference tasks.

In Section 2 we clarify notation, recall necessary theory, and outline the vanilla and hybrid SVGD algorithms. Section 3 contains the theoretical contributions, with proofs relegated to Appendix A. Numerical experiments are in Section 4 with additional experiments and details in Appendix B.

## 2 Background

### 2.1 Notation

Let $\mathcal{X} \subseteq \mathbb{R}^d$. Let $\pi$ denote the target probability density on $\mathcal{X}$ and let $\boldsymbol{s}_\pi(\boldsymbol{x}) = \nabla_{\boldsymbol{x}} \log \pi(\boldsymbol{x})$. We will often write $\pi$ for the corresponding measure. Assume that $\pi(\boldsymbol{x}) = e^{-V(\boldsymbol{x})}$ for some potential $V$. Let $\mathcal{P}(\mathcal{X})$ be the set of probability measures on $\mathcal{X}$ and let $\mathcal{P}_V(\mathcal{X})$ denote the subset where $\|\mu\|_{\mathcal{P}_V} := \int_{\mathcal{X}} (1+V(\boldsymbol{x})) d\mu(\boldsymbol{x}) < \infty$. For each $p \geq 1$, let $\mathcal{P}_p(\mathcal{X})$ denote the subset satisfying $\|\mu\|_{\mathcal{P}_p} := \int_{\mathcal{X}} \|\boldsymbol{x}\|^p d\mu(\boldsymbol{x}) < \infty$ and define the Wasserstein $p$-distance between the two measures $\mu, \nu \in \mathcal{P}_p(\mathcal{X})$ as $W_p(\mu, \nu) := \left(\inf_{\gamma \in \Gamma(\mu, \nu)} \int \|\boldsymbol{x} - \boldsymbol{y}\|^p d\gamma(\mu, \nu)\right)^{1/p}$, where $\Gamma(\mu, \nu)$ is the set of couplings between $\mu$ and $\nu$. Let $L_c^\infty(\mathcal{X})$ denote the set of probability densities bounded almost everywhere with compact support. Let $L^2(\mu)$ denote the set of functions that are square integrable with respect to the measure $\mu$. Given $\mu \in \mathcal{P}(\mathcal{X})$ and a smooth, invertible transform $\boldsymbol{T} : \mathcal{X} \to \mathcal{X}$, let $\boldsymbol{T}_{\#}\mu$ denote the pushforward measure of $\mu$ through $\boldsymbol{T}$. The KL divergence between two measures $\mu, \nu \in \mathcal{P}(\mathcal{X})$ is denoted by $\mathrm{KL}(\mu \parallel \nu)$.

### 2.2 Reproducing Kernel Hilbert Spaces

A function $k : \mathcal{X} \times \mathcal{X} \to \mathbb{R}$ is positive definite if $\sum_{i,j} a_i k(\boldsymbol{x}_i, \boldsymbol{x}_j) a_j > 0$ for any choice of $a_1, \ldots, a_d \in \mathbb{R}$ and $\boldsymbol{x}_1, \ldots, \boldsymbol{x}_d \in \mathcal{X}$. Given a Hilbert space $\mathcal{H}$ of functions $\phi : \mathcal{X} \to \mathbb{R}$, a function $k : \mathcal{X} \times \mathcal{X} \to \mathbb{R}$ is said to be a reproducing kernel for $\mathcal{H}$ if it satisfies the reproducing property, $\phi(\boldsymbol{x}) = \langle \phi, k(\boldsymbol{x}, \cdot) \rangle_{\mathcal{H}}$ for all $\phi \in \mathcal{H}$. A positive definite $k : \mathcal{X} \times \mathcal{X} \to \mathbb{R}$ admits a unique Hilbert space $\mathcal{H}$ of functions $\phi : \mathcal{X} \to \mathbb{R}$ for which the Dirac functionals $\delta_{\boldsymbol{x}} : \mathcal{H} \to \mathbb{R}, \delta_{\boldsymbol{x}} \phi = \phi(\boldsymbol{x})$ are all continuous and $k$ is a reproducing kernel. This Hilbert space is called the reproducing kernel Hilbert space (RKHS) of $k$ and it is equal to the closure of the span of $\{k(\boldsymbol{x}, \cdot) : \boldsymbol{x} \in \mathbb{R}\}$. Let $\mathcal{H}^d = \mathcal{H} \times \cdots \times \mathcal{H}$ denote the Hilbert space of functions $\phi : \mathcal{X} \to \mathbb{R}^d$ whose components are all in $\mathcal{H}$, and equip it with the usual inner product $\langle \phi, \psi \rangle_{\mathcal{H}^d} = \sum_{i=1}^d \langle \phi_i, \psi_i \rangle_{\mathcal{H}}$. Given two kernels $k_1, k_2 : \mathcal{X} \times \mathcal{X} \to \mathbb{R}$, let $\mathcal{H}_1, \mathcal{H}_2$ denote their respective RKHS. An important kernel used throughout this paper is the radial basis function (RBF) kernel $k_{\mathrm{RBF}}(\boldsymbol{x}, \boldsymbol{y}; h) := \exp(-\|\boldsymbol{x} - \boldsymbol{y}\|_2^2 / (2h))$ with bandwidth $h > 0$. For a thorough treatment of RKHS we refer the reader to Aronszajn (1950), Steinwart & Christmann (2008) and Berlinet & Thomas-Agnan (2011).

### 2.3 Stein Variational Gradient Descent

The key result from Liu & Wang (2016) identifies a transform $\boldsymbol{T} : \mathcal{X} \to \mathcal{X}$ that optimally decreases the KL divergence from an arbitrary probability measure to $\pi$. More precisely, let $\mathcal{H}$ be an RKHS with kernel $k : \mathcal{X} \times \mathcal{X} \to \mathbb{R}$ and consider transforms of the form $\boldsymbol{T}(\boldsymbol{x}) = \boldsymbol{x} + \epsilon \phi(\boldsymbol{x})$ where $\epsilon > 0$ and $\phi$ is in the unit ball $\{\phi \in \mathcal{H}^d : \|\phi\|_{\mathcal{H}^d} \leq 1\}$. The maximum value of

$$-\nabla_\epsilon \mathrm{KL}(\boldsymbol{T}_{\#}\mu \parallel \pi)|_{\epsilon=0} \tag{1}$$

occurs at $\phi_{\mu,\pi}^k / \|\phi_{\mu,\pi}^k\|_{\mathcal{H}^d}$, where

$$\phi_{\mu,\pi}^k(\cdot) := \mathbb{E}_{\boldsymbol{x} \sim \mu} [k(\boldsymbol{x}, \cdot) \boldsymbol{s}_\pi(\boldsymbol{x}) + \nabla_{\boldsymbol{x}} k(\boldsymbol{x}, \cdot)]. \tag{2}$$

When $\mu$ is an empirical distribution (i.e. a sum of Dirac measures), the expectation in (2) can be computed exactly by summing over the particles of each Dirac measure. Using this observation, the SVGD algorithm starts with an initial set of $N$ particles $(\boldsymbol{x}_0^i)_{i=1}^N$ and iteratively applies the transform $\boldsymbol{T}$ with (2) as the update direction. At each iteration $\ell$, this yields a set of particles $(\boldsymbol{x}_\ell^i)_{i=1}^N$ and a corresponding empirical distribution $\mu_\ell = \frac{1}{N} \sum_i \delta_{\boldsymbol{x}_\ell^i}$. This is captured in Algorithm 1. The intention is that after sufficiently many iterations, the set of particles will resemble samples from $\pi$ and expectations of the form $\mathbb{E}_{\boldsymbol{x} \sim \pi} h(\boldsymbol{x})$ can be approximated by $\mathbb{E}_{\boldsymbol{x} \sim \mu_\ell} h(\boldsymbol{x}) = \frac{1}{N} \sum_i h(\boldsymbol{x}_\ell^i)$. We also recall the definition of the kernelised Stein discrepancy (KSD) from Liu et al. (2016),

$$\mathbb{S}_k(\mu, \pi) := \mathbb{E}_{\boldsymbol{x}, \boldsymbol{y} \sim \mu} \left[ \boldsymbol{\delta}_{\pi,\mu}(\boldsymbol{x})^\top k(\boldsymbol{x}, \boldsymbol{y}) \boldsymbol{\delta}_{\pi,\mu}(\boldsymbol{x}) \right] \tag{3}$$

where $\boldsymbol{\delta}_{\pi,\mu} := \boldsymbol{s}_\pi(\boldsymbol{x}) - \boldsymbol{s}_\mu(\boldsymbol{x})$.

---

**Algorithm 1** Stein Variational Gradient Descent (Liu & Wang, 2016)

---

**Input:** A target probability distribution $\pi$, a kernel $k$, an initial set of particles $(\boldsymbol{x}_0^i)_{i=1}^N$ in $\mathcal{X}$, and a sequence of step sizes $(\epsilon_\ell)$.
**Output:** A set of particles $(\boldsymbol{x}^i)_{i=1}^N$ in $\mathcal{X}$ whose empirical distribution approximates $\pi$.
**for** iteration $\ell$ **do**

$$\boldsymbol{x}_{\ell+1}^i \leftarrow \boldsymbol{x}_\ell^i + \epsilon_\ell \hat{\phi}_{\mu_\ell,\pi}^*(\boldsymbol{x}_\ell^i), \quad \forall\ i = 1, \ldots, N$$

$$\hat{\phi}_{\mu_\ell,\pi}^*(\boldsymbol{x}) = \frac{1}{N} \sum_{j=1}^N \underbrace{k(\boldsymbol{x}_\ell^j, \boldsymbol{x}) \boldsymbol{s}_\pi(\boldsymbol{x}_\ell^j)}_{\text{driving force}} + \underbrace{\nabla_{\boldsymbol{x}_\ell^j} k(\boldsymbol{x}_\ell^j, \boldsymbol{x})}_{\text{repulsive force}} \tag{4}$$

---

**Algorithm 2** Hybrid Kernel Stein Variational Gradient Descent

---

**Input:** A target probability distribution $\pi$, two kernels $k_1$, $k_2$, an initial set of particles $(\boldsymbol{x}_0^i)_{i=1}^N$ in $\mathcal{X}$, and a sequence of step sizes $(\epsilon_\ell)$.
**Output:** A set of particles $(\boldsymbol{x}^i)_{i=1}^N$ in $\mathcal{X}$ whose empirical distribution approximates $\pi$.
**for** iteration $\ell$ **do**

$$\boldsymbol{x}_{\ell+1}^i \leftarrow \boldsymbol{x}_\ell^i + \epsilon_\ell \hat{\phi}_{\mu_\ell,\pi}^*(\boldsymbol{x}_\ell^i), \quad \forall i = 1, \ldots, N$$

$$\hat{\phi}_{\mu_\ell,\pi}^*(\boldsymbol{x}) = \frac{1}{N} \sum_{j=1}^N \underbrace{k_1(\boldsymbol{x}_\ell^j, \boldsymbol{x}) \boldsymbol{s}_\pi(\boldsymbol{x}_\ell^j)}_{\text{driving force}} + \underbrace{\nabla_{\boldsymbol{x}_\ell^j} k_2(\boldsymbol{x}_\ell^j, \boldsymbol{x})}_{\text{repulsive force}} \tag{5}$$

---

### 2.4 Hybrid Kernel Stein Variational Gradient Descent

The SVGD update in (4) contains two terms, each using the same kernel. The first term, often referred to as the driving term, uses the score function to move particles towards regions of high probability density, and the repulsive term prevents particles from collapsing at the modes. The h-SVGD variant proposed by D'Angelo et al. (2021) uses a different kernel in each term. Let $k_1$ denote the kernel that appears alongside the score function, and let $k_2$ denote the repulsive kernel. For the remainder of this paper, $k_1$ and $k_2$ will both be positive definite. We present h-SVGD in Algorithm 2.

## 3 Theoretical Results

### 3.1 Definitions and Assumptions

A function $f : \mathcal{X} \to \mathbb{R}$ is in the Stein class of $\pi$ if it is smooth and satisfies $\int_{\boldsymbol{x} \in \mathcal{X}} \nabla_{\boldsymbol{x}} (f(\boldsymbol{x})\pi(\boldsymbol{x}))\, d\boldsymbol{x} = 0$. A function $\boldsymbol{f} = (f_1, \ldots, f_d) : \mathcal{X} \to \mathbb{R}^d$ is in the Stein class of $\pi$ if each $f_i$ belongs to the Stein class of $\pi$. A kernel $k : \mathcal{X} \times \mathcal{X} \to \mathbb{R}$ is in the Stein class of $\pi$ if it has continuous second order partial derivatives and both $k(\boldsymbol{x}, \cdot)$ and $k(\cdot, \boldsymbol{y})$ are in the Stein class of $\pi$ for all $\boldsymbol{x}, \boldsymbol{y} \in \mathcal{X}$. The hybrid Stein operator acts on a pair of kernels $k_1, k_2 : \mathcal{X} \times \mathcal{X} \to \mathbb{R}$ by

$$\mathcal{S}_\pi \otimes (k_1, k_2)(\boldsymbol{x}, \cdot) := k_1(\boldsymbol{x}, \cdot)\boldsymbol{s}_\pi(\boldsymbol{x}) + \nabla_{\boldsymbol{x}} k_2(\boldsymbol{x}, \cdot),$$

provided $k_1$ and $k_2$ both belong to the Stein class of $\pi$. This reduces to the Stein operator defined in Liu et al. (2016) when $k_1 = k_2$. Motivated by the h-SVGD update in (5), define the update direction

$$\phi_{\mu,\pi}^{k_1,k_2}(\cdot) := \mathbb{E}_{\boldsymbol{x} \sim \mu} \left[ \mathcal{S}_\pi \otimes (k_1, k_2)(\cdot, \boldsymbol{x}) \right], \tag{6}$$

and write $\phi^* = \phi_{\mu,\pi}^{k_1,k_2} / \left\| \phi_{\mu,\pi}^{k_1,k_2}(\cdot) \right\|_{\mathcal{H}_1}$ for the normalised direction. Let $\boldsymbol{G}(\,\cdot\,; k_1, \mu, \pi) := \mathbb{E}_{\boldsymbol{x} \sim \mu} [k_1(\boldsymbol{x}, \cdot)\boldsymbol{s}_\pi(\boldsymbol{x})]$ and $\boldsymbol{R}(\,\cdot\,; k_2, \mu) := \mathbb{E}_{\boldsymbol{x} \sim \mu} [\nabla_{\boldsymbol{x}} k_2(\boldsymbol{x}, \cdot)]$ denote the driving (or gradient) term and the repulsive term respec-

tively. We can then write $\phi_{\mu,\pi}^{k_1,k_2}(\cdot) = \boldsymbol{G}(\,\cdot\,;k_1,\mu,\pi) + \boldsymbol{R}(\,\cdot\,;k_2,\mu)$. The update transform

$$\boldsymbol{T}_{\mu,\pi}^{k_1,k_2}(\boldsymbol{x}) = \boldsymbol{x} + \epsilon\phi_{\mu,\pi}^{k_1,k_2}(\boldsymbol{x}) \tag{7}$$

and the map $\Phi_\pi^{k_1,k_2} : \mu \mapsto \left(\boldsymbol{T}_{\mu,\pi}^{k_1,k_2}\right)_\# \mu$ characterise the h-SVGD dynamics. For each $\ell$, define

$$\mu_{\ell+1}^N := \Phi_\pi^{k_1,k_2}(\mu_\ell^N), \qquad\qquad \mu_{\ell+1}^\infty := \Phi_\pi^{k_1,k_2}(\mu_\ell^\infty), \tag{8}$$

where $\mu_0^N$ is the empirical measure of the initial particles $(\boldsymbol{x}_0^i)_{i=1}^N$ drawn i.i.d. from some $\mu_0^\infty$.

All technical assumptions required in the theorems throughout this section are detailed here for completeness. The first set of assumptions relate to the potential of the target distribution.

(**A**1) $V \in C^\infty(\mathcal{X})$, $V \geq 0$, and $\lim_{|\boldsymbol{x}|\to\infty} V(\boldsymbol{x}) = +\infty$.

(**A**2) There exist constants $C_V > 0$ and $q > 1$ such that for all $\boldsymbol{x}, \boldsymbol{y} \in \mathcal{X}$,

$$|V(\boldsymbol{x})|^q \leq C_V(1 + V(\boldsymbol{x}))$$

and

$$\sup_{\theta\in[0,1]} \left|\nabla^2 V(\theta\boldsymbol{x} + (1-\theta)\boldsymbol{y})\right|^q \leq C_V(1 + V(\boldsymbol{x}) + V(\boldsymbol{y})).$$

(**A**3) For any $\alpha, \beta > 0$, there exists a constant $C_{\alpha,\beta} > 0$ such that

$$(1 + |\boldsymbol{x}|)(|\nabla V(\boldsymbol{y})| + \left|\nabla^2 V(\boldsymbol{y})\right|) \leq C_{\alpha,\beta}(1 + V(\boldsymbol{x}))$$

whenever $|\boldsymbol{y}| \leq \alpha\,|\boldsymbol{x}| + \beta$.

(**A**4) The Hessian $H_V$ of $V$ is well-defined and satisfies $\|H_V\|_{\text{op}} \leq M$ for some $M > 0$.

Assumptions (**A**1), (**A**2) and (**A**3) are identical to those from Lu et al. (2019). Assumption (**A**4) is identical to Assumption (A2) from Korba et al. (2020), and Assumption 2.1 from Salim et al. (2022). Assumptions on the kernels are also required.

(**B**1) There exist symmetric functions $K_1, K_2 : \mathcal{X} \to \mathbb{R}$ such that $k_i(\boldsymbol{x}, \boldsymbol{y}) = K_i(\boldsymbol{x} - \boldsymbol{y})$ for $i = 1, 2$, $K_1$ is $C^2$ with bounded derivatives, and $K_2$ is $C^4$ with bounded derivatives. We use $B > 0$ as a bound for all derivatives in the proofs.

(**B**2) There exists a constant $D > 0$ such that both $k_1$ and $\nabla k_2$ are $D$-Lipschitz, and such that $\nabla V(\cdot)k_1(\cdot, \boldsymbol{z})$ is $D$-Lipschitz for each $\boldsymbol{z}$. That is,

$$|k_1(\boldsymbol{x}, \boldsymbol{x}') - k_1(\boldsymbol{y}, \boldsymbol{y}')| \leq D\left(\|\boldsymbol{x} - \boldsymbol{y}\|_2 + \|\boldsymbol{x}' - \boldsymbol{y}'\|_2\right),$$
$$\|\nabla_{\boldsymbol{x}} k_2(\boldsymbol{x}, \boldsymbol{x}') - \nabla_{\boldsymbol{y}} k_2(\boldsymbol{y}, \boldsymbol{y}')\| \leq D\left(\|\boldsymbol{x} - \boldsymbol{y}\|_2 + \|\boldsymbol{x}' - \boldsymbol{y}'\|_2\right),$$
$$\|\nabla V(\boldsymbol{x})k_1(\boldsymbol{x}, \boldsymbol{z}) - \nabla V(\boldsymbol{y})k_1(\boldsymbol{y}, \boldsymbol{z})\| \leq D\left(\|\boldsymbol{x} - \boldsymbol{y}\|_2\right)$$

for all $\boldsymbol{x}, \boldsymbol{x}', \boldsymbol{y}, \boldsymbol{y}', \boldsymbol{z} \in \mathcal{X}$.

Assumption (**B**1) is a slight relaxation of Assumption 2.1 from Lu et al. (2019), and contains Assumption 2.6 from Salim et al. (2022). The first two parts of Assumption (**B**2) are hybrid kernel versions of Assumption (B2) from Korba et al. (2020), and the third part of Assumption (**B**2) relaxes the restrictive Assumption (B1) from Korba et al. (2020).

### 3.2 Large Particle Limit

We begin our theoretical study with a result on the deviation of the empirical distributions of h-SVGD to the large particle limit. The single kernel version (Korba et al., 2020) of the following result uses an assumption that is quite restrictive. It requires $|V(\boldsymbol{x})| \leq C_V$ for some constant $C_V > 0$, which rules out even a Gaussian target distribution. We relax this with the third part of Assumption (**B2**) and provide an updated proof in the appendix.

**Proposition 3.1.** *Assume (**A1**), (**A4**), (**B1**) and (**B2**), and let $T > 0$. For any $0 \leq \ell \leq \frac{T}{\epsilon_\ell}$, there exists a constant $L$ depending on $k_1, k_2$ and $p$ such that*

$$\mathbb{E}\left[W_2^2(\mu_\ell^N, \mu_\ell^\infty)\right] \leq \frac{1}{2\sqrt{N}} \sqrt{\mathrm{var}(\mu_0^\infty)} e^{LT}(e^{2LT} - 1).$$

### 3.3 Hybrid Stein PDE

In the continuous time limit, equation (5) becomes a coupled system of differential equations,

$$\frac{d\boldsymbol{x}_i}{dt} = \frac{1}{N} \sum_{j=1}^N k_1(\boldsymbol{x}_j, \boldsymbol{x}_i) \boldsymbol{s}_\pi(\boldsymbol{x}_j) + \nabla_{\boldsymbol{x}_j} k_2(\boldsymbol{x}_j, \boldsymbol{x}_i) \tag{9}$$

for $i = 1, \ldots, N$. In the mean field limit, integration by parts and the identity $\nabla \rho = \rho \nabla \log \rho$ yields

$$\frac{d\boldsymbol{x}}{dt} = \int \left(k_1(\boldsymbol{x}', \boldsymbol{x}) \boldsymbol{s}_\pi(\boldsymbol{x}') + \nabla_{\boldsymbol{x}'} k_2(\boldsymbol{x}', \boldsymbol{x})\right) \rho(\boldsymbol{x}') d\boldsymbol{x}' \tag{10}$$

$$= \int k_1(\boldsymbol{x}', \boldsymbol{x}) \nabla \log \pi(\boldsymbol{x}') \rho(\boldsymbol{x}') d\boldsymbol{x}' - \int k_2(\boldsymbol{x}', \boldsymbol{x}) \nabla_{\boldsymbol{x}'} \rho(\boldsymbol{x}') d\boldsymbol{x}'$$

$$= T_{k_1, \rho}\left(\nabla \log \pi\right)(\boldsymbol{x}) - T_{k_2, \rho}\left(\nabla \log \rho\right)(\boldsymbol{x}), \tag{11}$$

where $T_{k,\rho} : L^2(\rho)^d \to \mathcal{H}_k^d$ is the Hilbert-Schmidt operator given by $(T_{k,\rho}f)(\boldsymbol{x}) = \int k(\boldsymbol{x}', \boldsymbol{x}) f(\boldsymbol{x}') \rho(\boldsymbol{x}') d\boldsymbol{x}'$ for a kernel $k$. Recalling that $V$ is the potential of $\pi$ and so $V = -\log \pi$, this mean field limit can be described by the hybrid Stein partial differential equation

$$\partial_t \rho_t = \nabla \cdot \left(\rho_t \left(K_1 * \nabla V \rho_t + K_2 * \nabla \rho_t\right)\right). \tag{12}$$

**Definition 3.2.** Given a probability measure $\nu$ on $\mathbb{R}^d$, a map $X(t, \boldsymbol{x}; \nu) : [0, \infty) \times \mathbb{R}^d \to \mathbb{R}^d$ that is $C^1$ with respect to $t$ and satisfies

$$\partial_t X(t, \boldsymbol{x}; \nu) = -\left(K_1 * (\nabla V \rho_t)\right)(X(t, \boldsymbol{x}; \nu)) - (\nabla K_2 * \rho_t)(X(t, \boldsymbol{x}; \nu))$$
$$\rho_t = X(t, \cdot, \nu)_\# \nu \tag{13}$$
$$X(0, \boldsymbol{x}; \nu) = \boldsymbol{x}$$

is called a mean field characteristic flow of (10) or of (12).

We now generalise Theorem 2.4 from Lu et al. (2019), ensuring the existence of a solution to the hybrid Stein PDE. First, define $Y := \{u \in C(\mathcal{X}, \mathcal{X}) : \sup_{\boldsymbol{x} \in \mathcal{X}} |u(\boldsymbol{x}) - \boldsymbol{x}| < \infty\}$ with $d_Y(u, v) = \sup_{\boldsymbol{x} \in \mathcal{X}} |u(\boldsymbol{x}) - v(\boldsymbol{x})|$ and note that $(Y, d_Y)$ is a complete metric space.

**Proposition 3.3.** *Assume (**A1**), (**A2**), (**A3**) and (**B1**), and let $T > 0$. Then there exists a unique solution $X(\cdot, \cdot, \nu) \in C^1([0, T]; Y)$ to (13) and the corresponding $\rho_t$ is a weak solution to (12) satisfying*

$$\|\rho_t\|_{\mathcal{P}_V} \leq \|\pi\|_{\mathcal{P}_V} \exp\left(C \min\left(\|\nabla K_1\|_\infty, \|\nabla K_2\|_\infty\right) t\right) \tag{14}$$

*for some constant $C > 0$ depending on $K_1$, $K_2$ and $V$.*

The second kernel enables a stronger bound than Theorem 2.4 from Lu et al. (2019) by careful modification of the telescoping section of the proof (see Appendix A and Equation (3.8)). In particular, ensuring that $\|\nabla K_1\|_\infty < \|\nabla K_2\|_\infty$ when choosing $K_2$ yields a stronger bound in (14) than if $K_1$ were used for both kernels. We remark that this bound describes regularity of the solution to the PDE, not a rate of convergence.

### 3.4 Kernelised Wasserstein Gradient Flow and Asymptotic Density: $k_2 = ck_1$

Zhuo et al. (2018) uncovered a correlation between dimension and the magnitude of the repulsive force $\boldsymbol{R}$, as defined at the beginning of Section 3.1. Under some technical conditions, for any $\alpha, \delta \in (0, 1)$, they show that with probability at least $1 - \delta$, SVGD under an RBF kernel yields $\|\boldsymbol{R}(\,\cdot\,; k, \mu)\|_\infty = O(d^{-\alpha})$. This suggests that simply scaling the repulsive force by $d^\alpha$ for some $\alpha \in (0, 1)$ should offset the decrease in $\|\boldsymbol{R}(\,\cdot\,; k_2, \mu)\|_\infty$ in high dimensions, thereby alleviating variance collapse at negligible additional computational cost. Scaling the repulsive kernel in this way corresponds to h-SVGD where $k_1$ is an RBF kernel and $k_2 = d^\alpha k_1$. This motivates our study of the h-SVGD gradient flow under the special case $k_2 = ck_1$.

In the case where $k = k_1 = k_2$, equations (10) and (12) describe a kernelised Wasserstein gradient flow of the form $\partial_t \rho_t = \nabla \cdot (\rho_t T_{k,\rho_t} \nabla_W \mathcal{F}(\rho_t))$, where

$$\mathcal{F}(\rho) = \mathrm{KL}(\rho \parallel \pi) = \mathbb{E}_{\boldsymbol{x} \sim \rho} \left[ \log \rho(\boldsymbol{x}) - \log \pi(\boldsymbol{x}) \right]$$

is the KL divergence functional (Liu, 2017). Recall that a functional derivative of $\mathcal{F}$ is a measurable function $\frac{\delta \mathcal{F}}{\delta \rho}(\rho)$ satisfying

$$\frac{d}{d\epsilon} \mathcal{F}(\rho + \epsilon \chi) \Big|_{\epsilon=0} = \int \frac{\delta \mathcal{F}}{\delta \rho}(\rho) d\chi$$

for all perturbations $\chi = \tilde{\rho} - \rho$ with $\tilde{\rho} \in L_c^\infty(\mathcal{X}) \cap \mathcal{P}(\mathcal{X})$ (Santambrogio, 2015, Definition 7.12). In particular $\frac{\delta \mathcal{F}}{\delta \rho} = \log \rho - \log \pi + 1$ up to a constant, and $\nabla_W \mathcal{F}(\rho) = \nabla \frac{\delta \mathcal{F}}{\delta \rho}(\rho) = \nabla \log \rho - \nabla \log \pi$ is the Wasserstein gradient of $\mathcal{F}$. Using (11) and the linearity of $T_{k,\rho}$, the corresponding Fokker-Planck equation is then

$$\partial_t \rho_t = \nabla \cdot (\rho_t T_{k,\rho_t} (\nabla \log \rho_t - \nabla \log \pi)) \tag{15}$$

where $\{\rho_t : t \geq 0\}$ is a curve of probability densities. The following result generalises this gradient flow interpretation to the case where $k_2 = ck_1$. Note that it applies to any positive definite kernel $k_1$.

**Proposition 3.4.** *Given a positive definite kernel $k_1$ and constant $c > 0$, let $k_2 = ck_1$. Then the mean field dynamics of h-SVGD describe a kernelised Wasserstein gradient flow on the functional*

$$\mathcal{F}(\rho) = \mathbb{E}_{\boldsymbol{x} \sim \rho} \left[ c \log \rho - \log \pi(\boldsymbol{x}) \right]. \tag{16}$$

*The corresponding continuity equation is*

$$\partial_t \rho_t = \nabla \cdot (\rho_t T_{k_1,\rho_t} (c \nabla \log \rho_t - \nabla \log \pi)). \tag{17}$$

Even in this simple hybrid kernel setting, the following result establishes that the limiting distribution $\rho^*$ of the mean field regime is not equal to the target distribution $\pi$.

**Corollary 3.5.** *If $k_2 = ck_1$ for some constant $c > 0$ where $k_1$ is a positive definite kernel, then the mean field h-SVGD has a fixed point $\rho^*(\boldsymbol{x}) \propto \pi(\boldsymbol{x})^{1/c}$.*

Although Corollary 3.5 applies to any target density $\pi$ satisfying (**A**1), it is insightful to consider a Gaussian target. If $\pi(\boldsymbol{x}) = \mathcal{N}(\boldsymbol{x}; \boldsymbol{\mu}, \boldsymbol{\Sigma})$, then $\rho^*(\boldsymbol{x}) = \mathcal{N}(\boldsymbol{x}; \boldsymbol{\mu}, c\boldsymbol{\Sigma})$. So scaling the repulsive kernel $k_2$ adjusts the variance of the target by the same factor. This supports the motivation that scaling the repulsive kernel should offset the variance underestimation in the finite particle regime in high dimensions at a negligible additional cost. This idea will be revisited in Section 4. Note that Corollary 3.5 extends the $c = 1$ case where SVGD converges to the target $\pi$ in the mean field limit (Liu & Wang, 2016).

We now generalise an existing result (Korba et al., 2020) that describes the dissipation of the KL divergence along the SVGD gradient flow. The result below describes the dissipation of the functional in (16) along the h-SVGD gradient flow, ensuring that the functional always decreases. It also describes the dissipation of the KL divergence with respect to the mean field limiting distribution $\rho^*$, which we emphasise is not equal to the target distribution $\pi$, as per Corollary 3.5.

**Proposition 3.6.** *Under the assumptions of Proposition 3.4,*

$$\frac{d}{dt}\mathcal{F}(\rho_t) = -c^2 \|T_{k_1,\rho_t}(\nabla \log \rho - \nabla \log \rho^*)\|_{\mathcal{H}_1^d}^2 + c \int \rho_t(\boldsymbol{x}) \frac{\partial}{\partial t} \log \rho_t(\boldsymbol{x}) d\boldsymbol{x}, \tag{18}$$

$$\frac{d}{dt}\mathrm{KL}(\rho_t \parallel \rho^*) = -c \|T_{k_1,\rho_t}(\nabla \log \rho - \nabla \log \rho^*)\|_{\mathcal{H}_1^d}^2 + \int \rho_t(\boldsymbol{x}) \frac{\partial}{\partial t} \log \rho_t(\boldsymbol{x}) d\boldsymbol{x}, \tag{19}$$

*where $\rho^*(\boldsymbol{x}) \propto \pi(\boldsymbol{x})^{1/c}$ is the mean field fixed point. Furthermore,*

$$\int \rho_t(\boldsymbol{x}) \frac{\partial}{\partial t} \log \rho_t(\boldsymbol{x}) d\boldsymbol{x} \le 0,$$

*so $\frac{d}{dt}\mathcal{F}(\rho_t) \le 0$ and $\frac{d}{dt}\mathrm{KL}(\rho \parallel \rho^*) \le 0$ for $t \ge 0$, and $\frac{d}{dt}\mathcal{F}(\rho^*) = 0$.*

The following descent lemma is adapted from Liu (2017) and provides a discrete time version of Proposition 3.6. We use $\mu_\ell$ to denote discrete time steps, as in (8), as opposed to $\rho_t$ for the continuous time analysis. We remark that other descent lemmas for SVGD have been proved (Korba et al., 2020; Salim et al., 2022).

**Proposition 3.7.** *Under the assumptions of Proposition 3.4, let $\sigma(\cdot)$ denote the spectral radius and set $\epsilon_\ell \le (2 \sup_{\boldsymbol{x}} \sigma(\nabla \phi^*(\boldsymbol{x}) + \nabla \phi^*(\boldsymbol{x})^\top))^{-1}$. Define $R := \sup_{\boldsymbol{x}}\{\frac{1}{2}\|\nabla \log \pi\|_{\mathrm{Lip}} k_1(\boldsymbol{x},\boldsymbol{x}) + 2\nabla_{\boldsymbol{x},\boldsymbol{y}}k_1(\boldsymbol{x},\boldsymbol{x})\}$ where $\|f\|_{\mathrm{Lip}} := \sup_{x \ne y}\frac{|f(x)-f(y)|}{\|x-y\|_2}$ is the Lipschitz norm and $\nabla_{\boldsymbol{x},\boldsymbol{y}}k(\boldsymbol{x},\boldsymbol{x}) := \sum_i \partial_{x_i}\partial_{y_i}k(\boldsymbol{x},\boldsymbol{y})|_{\boldsymbol{x}=\boldsymbol{y}}$. Then*

$$\frac{1}{\epsilon_\ell}\left(\mathcal{F}(\mu_{\ell+1}^\infty) - \mathcal{F}(\mu_\ell^\infty)\right) \le -c(1 - \epsilon_\ell R)\, \mathbb{S}(\mu_\ell, \rho^*).$$

### 3.5 Kernelised Wasserstein Gradient Flow: The General Case

In this section, we present a generalisation of Proposition 3.4 and discuss some difficulties in finding kernels that satisfy the required conditions. For ease of presentation, we restrict our attention to $d = 1$. Throughout, we assume (**B**1) and that any required Fourier transforms exist. Define the function $r : \mathcal{X} \to \mathbb{R}$ by

$$r(x; \rho) := \mathscr{F}^{-1}\left(\frac{\mathscr{F}(K_2)}{\mathscr{F}(K_1)} \cdot \mathscr{F}(\nabla \rho)\right)(x) \tag{20}$$

where $\mathscr{F}$ denotes the Fourier transform (see proof in Appendix A), and let $R : \mathcal{X} \to \mathbb{R}$ be a function satisfying $\nabla R(x; \rho) = r(x; \rho)/\rho(x)$.

**Proposition 3.8.** *Assume that both $r$ and $R$ exist. Then the corresponding continuity equation is*

$$\partial_t \rho_t = \nabla \cdot \left(\rho_t T_{k_1,\rho_t}\left(\nabla R(\cdot\,; \rho_t) - \nabla \log \pi\right)\right). \tag{21}$$

*If in addition*

$$\int \frac{\partial}{\partial \epsilon} R(x; \rho + \epsilon\chi) d(\rho + \epsilon\chi)(x)\bigg|_{\epsilon=0} = 0 \tag{22}$$

*for any measure $\chi = \tilde{\rho} - \rho$ with $\tilde{\rho} \in L_c^\infty(\mathcal{X}) \cap \mathcal{P}(\mathcal{X})$, then the mean field dynamics of h-SVGD describe a kernelised Wasserstein gradient flow on the functional*

$$\mathcal{F}(\rho) = \mathbb{E}_{x \sim \rho}\left[R(x; \rho) - \log \pi(x)\right]. \tag{23}$$

Note that $k_2 = ck_1$ implies $\mathscr{F}(K_2)/\mathscr{F}(K_1) = c$ and so $r(x; \rho) = c\nabla\rho(x)$. Therefore, $R(x; \rho) = c\log\rho(x)$, and so (21) reduces to (17). Furthermore, the left hand side of (22) is equal to $c\int d\chi(\boldsymbol{x}) = c\int d\tilde{\rho}(x) - c\int d\rho(x)$, which is zero because $\tilde{\rho}, \rho$ are both probability measures. So (22) is satisfied in this special case. We remark that verifying (22) remains a challenge in general hybrid kernel settings. However, under this assumption, the mean field dynamics of h-SVGD converges to a distribution that is not the target distribution.

**Proposition 3.9.** *If $k_1 \ne k_2$, both $r$ and $R$ exist, and (22) is satisfied, then any fixed point $\rho^*$ of the gradient flow in Proposition (3.8) is not equal to $\pi$.*

The example below demonstrates some behaviour of SVGD in the general hybrid kernel setting. In particular, the form of the target $\pi$ and the mean field steady state $\rho^*$ can look quite different.

**Example 3.10.** *Let $h_1, h_2, \sigma > 0$ and assume that $\Delta h := h_2 - h_1 \neq 0$. Let $k_i(x,y) = k_{\mathrm{RBF}}(x,y;h_i)$ for $i = 1, 2$, and let $\pi(x) \propto \exp(-\alpha \exp(\frac{x^2}{2\beta}))$ be the target on $\mathcal{X} = \mathbb{R}$ where*

$$\alpha = \sqrt{\frac{h_2}{h_1}} \cdot \sqrt{\frac{\sigma^2}{\sigma^2 + \Delta h}} \cdot \frac{\sigma^2}{\Delta h}, \qquad\qquad \beta = \frac{\sigma^2(\sigma^2 + \Delta h)}{\Delta h}.$$

*Then $\rho^*(x) = \mathcal{N}(x; 0, \sigma^2)$ is a fixed point of the h-SVGD mean field dynamics.*

We briefly remark that the right hand side of Equation (20) exists in this setting of two RBF kernels with bandwidths $h_2 > h_1$ provided $\rho$ decays sufficiently fast. In particular,

$$\frac{\mathscr{F}(K_2)(\omega)}{\mathscr{F}(K_1)(\omega)} \cdot \mathscr{F}(\nabla\rho)(\omega) = i2\pi\omega\sqrt{\frac{h_2}{h_1}} \cdot \exp\left(-2\pi^2(h_2 - h_1)\omega^2\right)\mathscr{F}(\rho)(\omega).$$

We also remark that Section 4 focuses on experimental results in the $k_2 = ck_1$ case due to its capacity to improve variance estimation. We leave a more detailed study of the general hybrid setting for future work.

## 4 Experiments

Although Corollary 3.5 shows that h-SVGD does not converge to the target distribution, we demonstrate in this section that it has the ability to improve variance estimation when compared to SVGD. Furthermore, it does this at no extra computational cost, and without any assumptions on the structure of the posterior, as is common in other SVGD variants that alleviate variance collapse. We measure variance collapse using dimension averaged marginal variance (DAMV), $\frac{1}{d}\sum_{j=1}^{d} \mathrm{Var}_j\left(\{\boldsymbol{x}_i\}_{i=1}^{N}\right)$, as is standard in the literature (Zhuo et al., 2018; Ba et al., 2019; 2021; Gong et al., 2021). Further details on the the experimental details can be found in Appendix B.[1]

Before describing the first numerical example, we reiterate the intuition behind the ability of h-SVGD to improve variance estimation. In high dimensions, with a finite number of particles, SVGD is well known to suffer from variance collapse (Wang et al., 2018; Ba et al., 2019). Indeed, for any $\alpha, \delta \in (0, 1)$, the repulsive force of SVGD under an RBF kernel decreases as $\|\boldsymbol{R}(\,\cdot\,; k, \mu)\|_{\infty} = O(d^{-\alpha})$ with probability at least $1 - \delta$ (Zhuo et al., 2018). By Corollary 3.5, under a Gaussian target distribution, a repulsive kernel $k_2 = ck_1$ with $c > 0$ will scale the variance of the limiting distribution by $c$ in the infinite particle limit. So scaling the repulsive kernel by $c = d^\alpha$ for some $\alpha \in (0, 1)$ should compensate for the decrease in $\|\boldsymbol{R}(\,\cdot\,; k_2, \mu)\|_{\infty}$, and as a consequence offset the variance collapse in high dimensions in the finite particle regime. The factor $\alpha$ should be tuned and we expect that a suitable choice of $\alpha$ will depend on the number of particles $N$, the target distribution $\pi$ and the kernel $k_1$. We leave a more detailed theoretical analysis guiding the choice of $\alpha$ for future work. In the experiments that follow, we found that $\alpha = 0.5$ provided a good balance between variance estimation and inference.

### 4.1 Multivariate Gaussian Mixture

In this first example, we sample from a high-dimensional mixture of Gaussians with 10 equally weighted components. The mean vectors are randomly initialised from $\mathcal{N}(0, \boldsymbol{I}_d)$, and the covariance matrices are $c\boldsymbol{I}_d$, where $c > 0$ is a factor that makes the DAMV equal to 1. We repeat this for dimensions up to $d = 1000$ at intervals of 100. We choose to sample $N = 50$ particles in order to demonstrate the performance of h-SVGD when $d$ is much greater than $N$, as is often the case in high dimensional Bayesian inference. Particles are initialised from $\mathcal{N}(0, \boldsymbol{I}_d)$ and each SVGD variant is run for 2000 iterations with an initial step size of $\epsilon = 0.01$, adapted using AdaGrad. We run SVGD and sliced SVGD (SSVGD) (Gong et al., 2021) with the RBF kernel $k_{\mathrm{RBF}}$ and compare against h-SVGD with kernels $k_1 = k_{\mathrm{RBF}}$ and $k_2 = \sqrt{d} \cdot k_{\mathrm{RBF}}$. All algorithms use $h = \mathrm{med}^2/\log(N)$ as the bandwidth, where med is the median pairwise distance between particles (Liu

---

[1]The python code for reproducing these experiments is available at `https://github.com/anson-macdonald-unsw/h-SVGD`

& Wang, 2016). For SSVGD, there is one bandwidth per dimension, so the median distances are computed along each projection, as described in Gong et al. (2021). We also compute the energy distance between the samples generated by each method and the target distribution as described in Székely & Rizzo (2013). All configurations are run 10 times with a different initialisation and results for each configuration are averaged.

Figure 1 demonstrates that h-SVGD provides an uplift in marginal variance estimation and an improved energy distance when compared to SVGD. Although SSVGD estimates the variance fairly consistently as the dimension increases, Figure 1 shows that this comes at a significant increase in runtime, whereas there is no noticeable difference between the runtimes of SVGD and h-SVGD. Furthermore, h-SVGD outperforms SSVGD in terms of the energy distance. Figure 2 shows projections of the $N = 50$ particles onto the first two two dimensions along with a contour plot of the marginal density along those two dimensions. This visual representation emphasises the extent to which both h-SVGD and SSVGD offset variance collapse in the finite particle setting. Further figures are included in Appendix B.

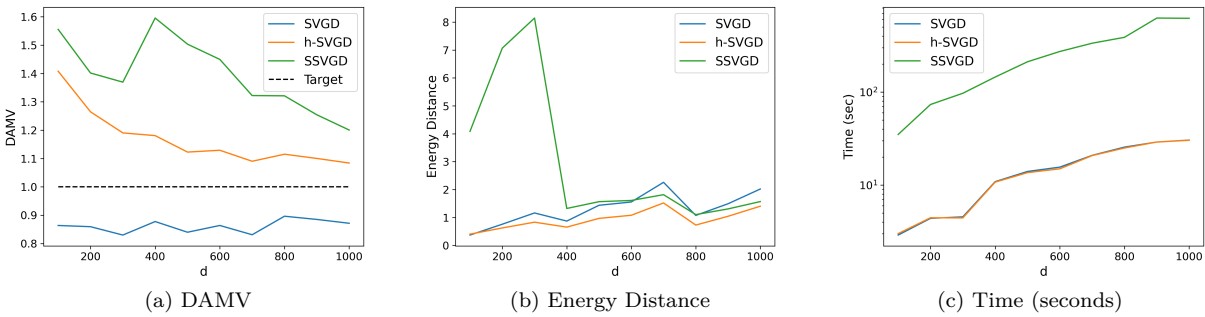

| (a) DAMV | (b) Energy Distance | (c) Time (seconds) |

Figure 1: DAMV, energy distance, and runtime of different SVGD variants sampling from high-dimensional Gaussian mixtures.

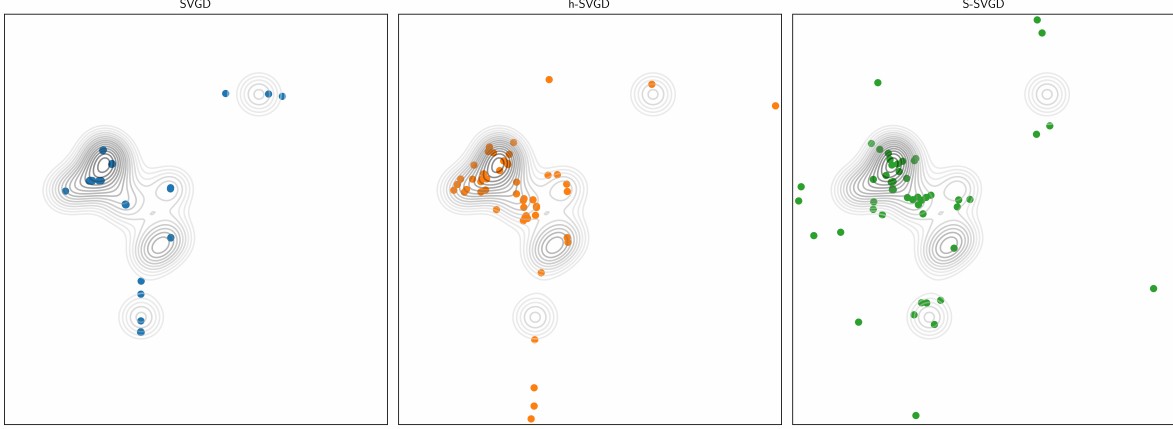

Figure 2: Projections onto the first two dimensions of particles sampled from a high-dimensional Gaussian mixture using different SVGD variants. The contour plot of the corresponding marginal density is overlaid.

## 4.2 Bayesian Neural Network

In this section, we sample weights from a Bayesian neural network (BNN). Aside from scaling the repulsive kernel by $\sqrt{d}$, as described in the previous experiment, our setup is identical to Liu & Wang (2016). For completeness, details are included in Appendix B. Using the no-U-turn sampler (NUTS) (Hoffman et al., 2014) with 10 parallel chains, we generated 10 000 ground truth samples for 8 of the 10 datasets. The Protein and Year datasets were large enough to make NUTS prohibitively slow. We then use these ground

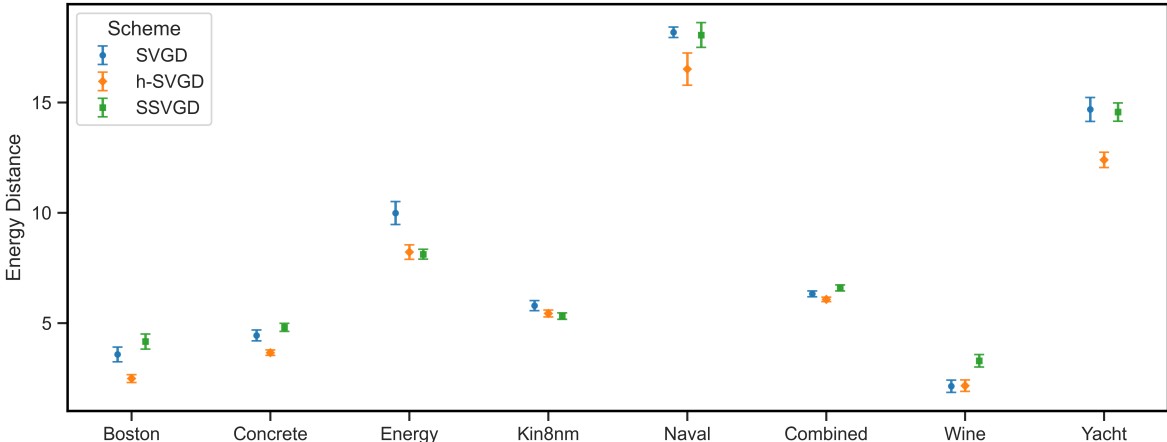

Figure 3: Energy distance of BNN samples generated by each SVGD variant as compared against ground truth samples generated by NUTS. Ranges indicate one standard deviation above and below the mean over repeated trials.

truth samples to compute the energy distance (Székely & Rizzo, 2013) of samples generated by each variant. Table 1 shows that with no additional computational cost, the problem of variance collapse is mitigated under h-SVGD through an increased DAMV. Table 2 demonstrates that it remains competitive at inference through improved test log-likelihood (LL) for all datasets and improved root mean squared error (RMSE) for all but one dataset. Appendix B includes a comparison of the same metrics between h-SVGD and SSVGD to emphasise that the alleviation of variance collapse under h-SVGD comes at a lower cost. Figure 3 shows that across all datasets except Wine, the samples generated by h-SVGD are closer to the ground truth than those generated by SVGD, as measured by the energy distance. The h-SVGD samples are also closer to the ground truth than the SSVGD samples for all but two datasets.

Table 1: DAMV and runtime in seconds of SVGD and h-SVGD. Ranges indicate one standard deviation above and below the mean over repeated trials. NUTS runtime is included for those datasets where it was computationally feasible.

| | DAMV | | Runtime (seconds) | | |
|---|---|---|---|---|---|
| **Dataset** | **SVGD** | **h-SVGD** | **SVGD** | **h-SVGD** | **NUTS** |
| Boston | $0.051 \pm 0.011$ | $\mathbf{0.112 \pm 0.015}$ | $28.9 \pm 1.4$ | $\mathbf{28.6 \pm 0.8}$ | 133 |
| Concrete | $0.084 \pm 0.010$ | $\mathbf{0.120 \pm 0.010}$ | $\mathbf{28.7 \pm 1.3}$ | $28.8 \pm 1.8$ | 189 |
| Energy | $0.065 \pm 0.015$ | $\mathbf{0.154 \pm 0.023}$ | $\mathbf{30.8 \pm 2.8}$ | $30.9 \pm 2.5$ | 167 |
| Kin8nm | $\mathbf{0.105 \pm 0.003}$ | $0.102 \pm 0.003$ | $\mathbf{35.9 \pm 1.3}$ | $36.2 \pm 1.3$ | 1 748 |
| Naval | $0.059 \pm 0.004$ | $\mathbf{0.091 \pm 0.019}$ | $33.4 \pm 0.8$ | $\mathbf{33.1 \pm 1.5}$ | 51 |
| Combined | $0.128 \pm 0.008$ | $\mathbf{0.145 \pm 0.007}$ | $\mathbf{36.3 \pm 1.5}$ | $36.8 \pm 2.7$ | 954 |
| Protein | $\mathbf{0.089 \pm 0.001}$ | $0.087 \pm 0.001$ | $\mathbf{72.7 \pm 1.0}$ | $72.8 \pm 1.7$ | NA |
| Wine | $0.068 \pm 0.005$ | $\mathbf{0.090 \pm 0.003}$ | $\mathbf{29.5 \pm 1.4}$ | $29.8 \pm 1.1$ | 393 |
| Yacht | $0.060 \pm 0.020$ | $\mathbf{0.194 \pm 0.034}$ | $\mathbf{29.3 \pm 0.4}$ | $29.6 \pm 0.4$ | 83 |
| Year | $0.011 \pm 0.000$ | $\mathbf{0.012 \pm 0.000}$ | $592 \pm 24$ | $\mathbf{563 \pm 23}$ | NA |

## 5  Conclusion

In this paper we have developed the mean field theory of h-SVGD by proving the existence of a solution to the hybrid Stein PDE and identifying it as a gradient flow on a functional other than the KL divergence. We characterised the mean field fixed point for the special case $k_2 = ck_1$ and demonstrated that h-SVGD does not converge to the target in the mean field limit unless $c = 1$. This suggests that h-SVGD may

Table 2: Average RMSE and LL of SGVD and h-SVGD evaluated on the test dataset. Ranges indicate one standard deviation above and below the mean over repeated trials.

| | Test RMSE | | Test LL | |
|---|---|---|---|---|
| **Dataset** | **SVGD** | **h-SVGD** | **SVGD** | **h-SVGD** |
| Boston | $3.094 \pm 0.579$ | **$3.034 \pm 0.587$** | $-2.123 \pm 0.116$ | **$-1.959 \pm 0.148$** |
| Concrete | $5.857 \pm 0.468$ | **$5.384 \pm 0.504$** | $-2.616 \pm 0.099$ | **$-2.499 \pm 0.128$** |
| Energy | $1.528 \pm 0.169$ | **$1.157 \pm 0.138$** | $-1.702 \pm 0.094$ | **$-1.072 \pm 0.096$** |
| Kin8nm | $0.124 \pm 0.005$ | **$0.100 \pm 0.003$** | $-1.293 \pm 0.108$ | **$-0.314 \pm 0.104$** |
| Naval | $0.006 \pm 0.000$ | **$0.004 \pm 0.000$** | $-1.353 \pm 0.161$ | **$-0.404 \pm 0.140$** |
| Combined | $4.105 \pm 0.220$ | **$4.072 \pm 0.220$** | $-2.459 \pm 0.051$ | **$-2.367 \pm 0.050$** |
| Protein | $4.791 \pm 0.025$ | **$4.679 \pm 0.025$** | $-2.633 \pm 0.035$ | **$-2.511 \pm 0.023$** |
| Wine | $0.637 \pm 0.044$ | **$0.631 \pm 0.045$** | $-1.463 \pm 0.120$ | **$-0.819 \pm 0.080$** |
| Yacht | **$1.677 \pm 0.571$** | $1.886 \pm 0.664$ | $-1.587 \pm 0.120$ | **$-1.045 \pm 0.160$** |
| Year | $8.882 \pm 0.043$ | **$8.804 \pm 0.039$** | $-2.908 \pm 0.031$ | **$-2.890 \pm 0.019$** |

not converge to the target more generally, even in its continuous time, mean-field form. We provided a result on the dissipation of the new functional, as well as a discrete time version, otherwise known as a descent lemma. We also highlighted the complexities of the gradient flow in the general hybrid kernel setting. Despite non-convergence to the target, experimental results demonstrated that h-SVGD can alleviate variance collapse in high dimensions observed in finite-particle SVGD at a much lower cost than other SVGD variants. We also showed that h-SVGD maintains its performance on high dimensional inference tasks, whilst improving variance estimation without the additional computational cost required of other SVGD variants. One interesting direction for future research is to find a principled method of scaling the repulsive kernel. Another avenue is to further develop the theory of h-SVGD in the general hybrid kernel setting. It would also be interesting to incorporate tempered target densities, as considered in (Chehab et al., 2024).

### Acknowledgments

SP gratefully acknowledges funding from the Eva Mayr-Stihl foundation.

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

# A  Proofs

**Lemma A.1.** *Under assumptions (**A**1), (**A**4), (**B**1), (**B**2) the map*

$$(\boldsymbol{z}, \mu) \mapsto E(\boldsymbol{z}, \mu) := \int_{\mathcal{X}} -k_1(\boldsymbol{x}, \boldsymbol{z}) \nabla V(\boldsymbol{x}) + \nabla_{\boldsymbol{x}} k_2(\boldsymbol{x}, \boldsymbol{z}) d\mu(\boldsymbol{x})$$

*is L-Lipschitz. That is,*

$$\|E(\boldsymbol{z}, \mu) - E(\boldsymbol{z}', \mu')\|_2 \le L(\|\boldsymbol{z} - \boldsymbol{z}'\|_2 + W_2(\mu, \mu'))$$

*where $L > 0$ depends on $k_1$, $k_2$ and $V$.*

*Proof.* Largely following the proof of Lemma 14 Korba et al. (2020), choosing an optimal coupling $\gamma$ of $\mu$ and $\mu'$,

$$
\begin{aligned}
\|E(\boldsymbol{z}, \mu) - E(\boldsymbol{z}', \mu')\|_2 \le \big|\big| &\mathbb{E}_\gamma \left[ \nabla V(\boldsymbol{x})(k_1(\boldsymbol{x}, \boldsymbol{z}) - k_1(\boldsymbol{x}', \boldsymbol{z}')) \right] \\
&+ \mathbb{E}_\gamma \left[ (\nabla V(\boldsymbol{x}') - \nabla V(\boldsymbol{x}))k_1(\boldsymbol{x}', \boldsymbol{z}') \right] \\
&+ \mathbb{E}_\gamma \left[ \nabla_{\boldsymbol{x}} k_2(\boldsymbol{x}, \boldsymbol{z}) - \nabla_{\boldsymbol{x}'} k_2(\boldsymbol{x}', \boldsymbol{z}') \right] \big|\big| \\
\le\ & D \mathbb{E}_\gamma \left[ \|\boldsymbol{x} - \boldsymbol{x}'\|_2 + \|\boldsymbol{z} - \boldsymbol{z}'\|_2 \right] \\
&+ BM \mathbb{E}_\gamma \left[ \|\boldsymbol{x} - \boldsymbol{x}'\|_2 \right] \\
&+ D \mathbb{E}_\gamma \left[ \|\boldsymbol{x} - \boldsymbol{x}'\|_2 + \|\boldsymbol{z} - \boldsymbol{z}'\|_2 \right] \\
\le\ & (2D + BM) \left( \|\boldsymbol{z} - \boldsymbol{z}'\|_2 + W_2(\mu, \mu') \right).
\end{aligned}
$$

Note that the second term is bounded using the relaxed Assumption (**B**2) and there is no need to require that $|V|$ is bounded by a constant. □

*Proof of Proposition 3.1.* This follows identically to the proof of Proposition 7 in Korba et al. (2020) with Lemma A.1 in place of Lemma 14. □

*Proof of Proposition 3.3.* The proof largely follows those of Theorems 2.4 and 3.2 in Lu et al. (2019) with some minor adjustments. Notably, after fixing $r > 0$ and defining

$$Y_r := \left\{ u \in Y : \sup_{\boldsymbol{x} \in \mathcal{X}} |u(\boldsymbol{x}) - \boldsymbol{x}| < r \right\}$$

and the complete metric space

$$
\begin{aligned}
S_r &:= C([0, T_0]; Y_r), \\
d_S(u, v) &:= \sup_{t \in [0, T_0]} d_Y(u(t), v(t))
\end{aligned}
$$

for some sufficiently small $T_0$ (to be determined later), the operator $\mathcal{G} : u(t, \cdot) \mapsto \mathcal{G}(u)(t, \cdot)$ must be modified to act on $u \in S_r$ via

$$
\begin{aligned}
\mathcal{G}(u)(t, \boldsymbol{x}) = \boldsymbol{x} &- \int_0^t \int_{\mathcal{X}} \nabla K_2(u(s, \boldsymbol{x}) - u(s, \boldsymbol{x}')) \nu(d\boldsymbol{x}') ds \\
&- \int_0^t \int_{\mathcal{X}} K_1(u(s, \boldsymbol{x}) - u(s, \boldsymbol{x}')) \nabla V(u(s, \boldsymbol{x}')) \nu(d\boldsymbol{x}') ds.
\end{aligned}
$$

Note that we use $\mathcal{G}$ instead of the $\mathcal{F}$ used in Lu et al. (2019) to avoid confusion between the functional $\mathcal{F}$ defined in (16). The same techniques of Lu et al. (2019) are sufficient to establish the required bounds to show that $\mathcal{G}$ is a contraction on $S_r$ for sufficiently small $T_0$. Note that Assumptions (**B**1), (**A**2) and (**A**3) are used to establish this. So the unique fixed point $X(\cdot, \cdot; \nu) \in S_r$ of $\mathcal{G}$ solves (13) in the interval $[0, T_0]$.

The $\min (\|\nabla K_1\|_\infty, \|\nabla K_2\|_\infty)$ term emerges because the telescoping in Equation (3.8) of Lu et al. (2019) can be performed with either kernel. The remainder of the proof follows Theorems 2.4 and 3.2 of Lu et al. (2019). □

*Proof of Proposition 3.4.* Following Definition 7.12 in Santambrogio (2015), a functional derivative of $\mathcal{F}$ is a measurable function $\frac{\delta \mathcal{F}}{\delta \rho}(\rho)$ satisfying

$$\frac{d}{d\epsilon}\mathcal{F}(\rho + \epsilon\chi)\bigg|_{\epsilon=0} = \int \frac{\delta\mathcal{F}}{\delta\rho}(\rho)d\chi$$

for all perturbations $\chi = \tilde{\rho} - \rho$ with $\tilde{\rho} \in L_c^\infty(\mathcal{X}) \cap \mathcal{P}(\mathcal{X})$. If it exists, $\frac{\delta\mathcal{F}}{\delta\rho}(\rho)$ is unique up to additive constants. We first compute

$$
\begin{aligned}
&\frac{d}{d\epsilon}\mathcal{F}(\rho + \epsilon\chi)\bigg|_{\epsilon=0}\\
&= \frac{d}{d\epsilon}\left(\int c\log(\rho(\boldsymbol{x}) + \epsilon\chi(\boldsymbol{x}))\rho(\boldsymbol{x})d\boldsymbol{x} + \epsilon\int c\log(\rho(\boldsymbol{x}) + \epsilon\chi(\boldsymbol{x}))\chi(\boldsymbol{x})d\boldsymbol{x}\right.\\
&\qquad\left. - \int \log\pi(\boldsymbol{x})\rho(\boldsymbol{x})d\boldsymbol{x} - \epsilon\int \log\pi(\boldsymbol{x})\chi(\boldsymbol{x})d\boldsymbol{x}\right)\bigg|_{\epsilon=0}\\
&= \left(\int c\frac{\chi(\boldsymbol{x})}{\rho(\boldsymbol{x}) + \epsilon\chi(\boldsymbol{x})}\rho(\boldsymbol{x})d\boldsymbol{x} + \int c\log(\rho(\boldsymbol{x}) + \epsilon\chi(\boldsymbol{x}))\chi(\boldsymbol{x})d\boldsymbol{x}\right.\\
&\qquad\left. + \epsilon\int c\frac{\chi(\boldsymbol{x})}{\rho(\boldsymbol{x}) + \epsilon\chi(\boldsymbol{x})}\chi(\boldsymbol{x})d\boldsymbol{x} - \int \log\pi(\boldsymbol{x})\chi(\boldsymbol{x})d\boldsymbol{x}\right)\bigg|_{\epsilon=0}\\
&= \int c\log\rho(\boldsymbol{x}) - \log\pi(\boldsymbol{x})\chi(\boldsymbol{x})d\boldsymbol{x} + c\int \chi(\boldsymbol{x})d\boldsymbol{x}.
\end{aligned}
\tag{24}
$$

Since $\rho, \tilde{\rho} \in \mathcal{P}(\mathcal{X})$, the final integral is zero. So the functional gradient of $\mathcal{F}$ is

$$\frac{\delta\mathcal{F}}{\delta\rho}(\rho) = c\log\rho - \log\pi.$$

Its Wasserstein gradient is then

$$\nabla_W\mathcal{F}(\rho) = c\nabla\log\rho - \nabla\log\pi.$$

Since $k_2 = ck_1$, equation (10) can be written as

$$
\begin{aligned}
\frac{d\boldsymbol{x}}{dt} &= \int \left(k_1(\boldsymbol{x}', \boldsymbol{x})\nabla_{\boldsymbol{x}'}\log\pi(\boldsymbol{x}') + c\nabla_{\boldsymbol{x}'}k_1(\boldsymbol{x}', \boldsymbol{x})\right)\rho(\boldsymbol{x}')d\boldsymbol{x}'\\
&= \int k_1(\boldsymbol{x}', \boldsymbol{x})\nabla_{\boldsymbol{x}'}\log\pi(\boldsymbol{x}')\rho(\boldsymbol{x}') - ck_1(\boldsymbol{x}', \boldsymbol{x})\nabla_{\boldsymbol{x}'}\rho(\boldsymbol{x}')d\boldsymbol{x}'\\
&= \int k_1(\boldsymbol{x}', \boldsymbol{x})\nabla_{\boldsymbol{x}'}\log\pi(\boldsymbol{x}')\rho(\boldsymbol{x}') - ck_1(\boldsymbol{x}', \boldsymbol{x})\nabla_{\boldsymbol{x}'}\log\rho(\boldsymbol{x}')\rho(\boldsymbol{x}')d\boldsymbol{x}'\\
&= T_{k_1,\rho}(\nabla\log\pi - c\nabla\log\rho)(\boldsymbol{x})\\
&= -T_{k_1,\rho}(\nabla_W\mathcal{F}(\rho))(\boldsymbol{x}).
\end{aligned}
$$

The continuity equation $\partial_t\rho_t + \nabla \cdot \left(\rho_t\frac{d\boldsymbol{x}}{dt}\right) = 0$ then becomes

$$
\begin{aligned}
\partial_t\rho_t &= \nabla \cdot (\rho_t T_{k_1,\rho}(\nabla_W\mathcal{F}(\rho))(\boldsymbol{x}))\\
&= \nabla \cdot (\rho_t T_{k_1,\rho_t}(c\nabla\log\rho_t - \nabla\log\pi)).
\end{aligned}
$$

$\square$

*Proof of Corollary (3.5).* A measure $\rho^*$ satisfying $c\nabla \log \rho^* = \nabla \log \pi$ will be a fixed point because this would imply $\nabla_W \mathcal{F}(\rho) = 0$. Solving this for $\rho^*$, we have

$$c\nabla \log \rho^*(\boldsymbol{x}) = \nabla \log \pi(\boldsymbol{x})$$
$$c \log \rho^*(\boldsymbol{x}) = \log \pi(\boldsymbol{x}) + A$$
$$\log(\rho^*(\boldsymbol{x})^c) = \log(e^A \pi(\boldsymbol{x}))$$
$$\rho^*(\boldsymbol{x})^c = e^A \pi(\boldsymbol{x})$$
$$\rho^*(\boldsymbol{x}) = e^{A/c} \pi(\boldsymbol{x})^{1/c}$$

for some $A \in \mathbb{R}$. $\qquad\square$

*Proof of Proposition 3.6.* Using (17), integration by parts, and the fact that $T_{k_1,\rho} : L^2(\rho_t)^d \to \mathcal{H}_1^d$ is the adjoint of the inclusion $\imath : \mathcal{H}_1^d \to L^2(\rho_t)^d$,

$$\frac{d}{dt}\mathcal{F}(\rho_t) = \frac{d}{dt}\int \rho_t(\boldsymbol{x})\left(c \log \rho_t(\boldsymbol{x}) - \log \pi(\boldsymbol{x})\right) d\boldsymbol{x}$$

$$= \int \left(c \log \rho_t(\boldsymbol{x}) - \log \pi(\boldsymbol{x})\right)\frac{\partial}{\partial t}\rho_t(\boldsymbol{x}) + c\rho_t(\boldsymbol{x})\frac{\partial}{\partial t}\log \rho_t(\boldsymbol{x})d\boldsymbol{x}$$

$$= \int \left(c \log \rho_t(\boldsymbol{x}) - \log \pi(\boldsymbol{x})\right)\nabla \cdot \left(\rho_t(\boldsymbol{x})T_{k_1,\rho_t}(c\nabla \log \rho_t(\boldsymbol{x}) - \nabla \log \pi)(\boldsymbol{x})\right) d\boldsymbol{x}$$
$$+ \int c\rho_t(\boldsymbol{x})\frac{\partial}{\partial t}\log \rho_t(\boldsymbol{x})d\boldsymbol{x}$$

$$= -\int \nabla \left(c \log \rho_t(\boldsymbol{x}) - \log \pi(\boldsymbol{x})\right)\cdot T_{k_1,\rho_t}(c\nabla \log \rho_t - \nabla \log \pi)(\boldsymbol{x})\rho_t(\boldsymbol{x})d\boldsymbol{x}$$
$$+ \int c\rho_t(\boldsymbol{x})\frac{\partial}{\partial t}\log \rho_t(\boldsymbol{x})d\boldsymbol{x}$$

$$= -\langle c\nabla \log \rho_t - \nabla \log \pi, T_{k_1,\rho_t}(c\nabla \log \rho_t - \nabla \log \pi)\rangle_{L^2(\rho_t)^d}$$
$$+ \int c\rho_t(\boldsymbol{x})\frac{\partial}{\partial t}\log \rho_t(\boldsymbol{x})d\boldsymbol{x}$$

$$= -\|T_{k_1,\rho_t}(c\nabla \log \rho_t - \nabla \log \pi)\|_{\mathcal{H}_1^d}^2 + c\int \rho_t(\boldsymbol{x})\frac{\partial}{\partial t}\log \rho_t(\boldsymbol{x})d\boldsymbol{x}$$

$$= -c^2\|T_{k_1,\rho_t}(\nabla \log \rho_t - \nabla \log \rho^*)\|_{\mathcal{H}_1^d}^2 + c\int \rho_t(\boldsymbol{x})\frac{\partial}{\partial t}\log \rho_t(\boldsymbol{x})d\boldsymbol{x}. \tag{25}$$

Also,

$$\mathcal{F}(\rho_t) = \mathbb{E}_{\boldsymbol{x}\sim\rho_t}\left[c \log \rho_t(\boldsymbol{x}) - \log \pi(\boldsymbol{x})\right]$$
$$= \mathbb{E}_{\boldsymbol{x}\sim\rho_t}\left[c \log \rho_t(\boldsymbol{x}) - \log\left(A\rho^*(\boldsymbol{x})^c\right)\right]$$
$$= c\mathbb{E}_{\boldsymbol{x}\sim\rho_t}\left[\log \rho_t(\boldsymbol{x}) - \log \rho^*(\boldsymbol{x})\right] + \log(A)$$
$$= c\mathrm{KL}(\rho_t \parallel \rho^*) + \log(A) \tag{26}$$

for some $A \in \mathbb{R}$, we have $\frac{1}{c}\frac{d}{dt}\mathcal{F}(\rho_t) = \frac{d}{dt}\mathrm{KL}(\rho_t \parallel \rho^*)$. Therefore,

$$\frac{d}{dt}\mathrm{KL}(\rho_t \parallel \rho^*) = -\frac{1}{c}\|T_{k_1,\rho_t}(c\nabla \log \rho_t - \nabla \log \pi)\|_{\mathcal{H}_1^d}^2 + \int \rho_t(\boldsymbol{x})\frac{\partial}{\partial t}\log \rho_t(\boldsymbol{x})d\boldsymbol{x}$$

$$= -c\left\|T_{k_1,\rho_t}(\nabla \log \rho_t - \nabla \log \pi^{1/c})\right\|_{\mathcal{H}_1^d}^2 + \int \rho_t(\boldsymbol{x})\frac{\partial}{\partial t}\log \rho_t(\boldsymbol{x})d\boldsymbol{x}$$

$$= -c\|T_{k_1,\rho_t}(\nabla \log \rho_t - \nabla \log \rho^*)\|_{\mathcal{H}_1^d}^2 + \int \rho_t(\boldsymbol{x})\frac{\partial}{\partial t}\log \rho_t(\boldsymbol{x})d\boldsymbol{x}.$$

To simplify notation in the calculations below, set $u = T_{k_1,\rho_t}(c\nabla \log \rho_t)$ and $v = T_{k_1,\rho_t}(\nabla \log \pi)$. Recall the identity

$$\langle u, u - v \rangle = \frac{1}{2}\left(\|u\|^2 + \|u - v\|^2 - \|v\|^2\right). \tag{27}$$

Now we apply the multivariate chain rule to the remainder term along with the fact about the adjoint of $T_{k_1,\rho_t}$, the identity in (27), and the triangle inequality. This gives

$$\int \rho_t(\boldsymbol{x})\frac{\partial}{\partial t}\log \rho_t(\boldsymbol{x})d\boldsymbol{x} = \int \rho_t(\boldsymbol{x})\nabla \log \rho_t(\boldsymbol{x}) \cdot \frac{d\boldsymbol{x}}{dt}d\boldsymbol{x}$$

$$= -\int \rho_t(\boldsymbol{x})\nabla \log \rho_t(\boldsymbol{x}) \cdot T_{k_1,\rho_t}(c\nabla \log \rho_t - \nabla \log \pi)(\boldsymbol{x})d\boldsymbol{x} \tag{28}$$

$$= -\langle \nabla \log \rho_t, T_{k_1,\rho_t}(c\nabla \log \rho_t - \nabla \log \pi)\rangle_{L^2(\rho_t)^d}$$

$$= -\langle T_{k_1,\rho_t}(\nabla \log \rho_t), T_{k_1,\rho_t}(c\nabla \log \rho_t - \nabla \log \pi)\rangle_{\mathcal{H}_1^d}$$

$$= -\frac{1}{c}\langle u, u - v\rangle_{\mathcal{H}_1^d}$$

$$= \frac{1}{2c}\left(-\|u\|^2_{\mathcal{H}_1^d} + \|v\|^2_{\mathcal{H}_1^d} - \|u - v\|^2_{\mathcal{H}_1^d}\right)$$

$$\leq \frac{1}{2c}\left(-\|u\|^2_{\mathcal{H}_1^d} + \|u - v\|^2_{\mathcal{H}_1^d} + \|u\|^2_{\mathcal{H}_1^d} - \|u - v\|^2_{\mathcal{H}_1^d}\right)$$

$$= 0.$$

The final statement that $\frac{d}{dt}\mathcal{F}(\rho^*) = 0$ follows from equations (25) and (28) by observing that $c\nabla \log \rho^* - \nabla \log \pi = 0$. $\qquad\square$

*Proof of Proposition 3.7.* This follows from applying Theorem 3.3 Liu (2017) with $\rho^*$ instead of the target, then substituting in (26). $\qquad\square$

*Proof of Proposition 3.8.* We first define the Fourier transform of a real-valued function $f$ to be the function $\mathscr{F}(f)$ defined by

$$\mathscr{F}(f)(\omega) = \int f(x)\exp(-i2\pi\omega)dx,$$

given that the integral exists. Similarly, the inverse Fourier transform of a real-valued function $F$ is the function $\mathscr{F}^{-1}(F)$ defined by

$$\mathscr{F}^{-1}(F)(x) = \int F(\omega)\exp(i2\pi\omega x)d\omega,$$

where the integral exists. To recover the continuity equation, apply the convolution theorem to (20)

$$\mathscr{F}^{-1}\left(\frac{\mathscr{F}(K_2)}{\mathscr{F}(K_1)} \cdot \mathscr{F}(\nabla\rho)\right)(x) = r(x;\rho)$$

$$\mathscr{F}(K_2)(\omega) \cdot \mathscr{F}(\nabla\rho)(\omega) = \mathscr{F}(K_1)(\omega) \cdot \mathscr{F}(r)(\omega)$$

$$\mathscr{F}(K_2 * \nabla\rho)(\omega) = \mathscr{F}(K_1 * r)(\omega)$$

$$(K_2 * \nabla\rho)(x) = (K_1 * r)(x)$$

$$\int K_2(x - y)\nabla \log \rho(y)\rho(y)dy = \int K_1(x - y)\frac{r(y)}{\rho(y)}\rho(y)dy$$

$$(T_{k_2,\rho}\nabla \log \rho)(x) = (T_{k_1,\rho}(r/\rho))(x)$$

Equation (10) can now be rewritten as

$$
\begin{aligned}
\frac{dx}{dt} &= \int \left( k_1(x',x)\nabla_{x'}\log\pi(x') + \nabla_{x'}k_2(x',x) \right)\rho(x')dx' \\
&= \int k_1(x',x)\nabla_{x'}\log\pi(x')\rho(x') - k_2(x',x)\nabla_{x'}\rho(x')dx' \\
&= \int \left( k_1(x',x)\nabla_{x'}\log\pi(x') - k_2(x',x)\nabla_{x'}\log\rho(x') \right)\rho(x')dx' \\
&= \left( T_{k_1,\rho}\nabla\log\pi \right)(x) - \left( T_{k_2,\rho}\nabla\log\rho \right)(x) \\
&= \left( T_{k_1,\rho}\left( \nabla\log\pi - \nabla R(\,\cdot\,;\rho) \right) \right)(x),
\end{aligned}
$$

where $\nabla R(x;\rho) := \frac{r(x;\rho)}{\rho(x)}$, yielding the continuity equation (21).

As in the proof of Proposition 3.4, let $\chi = \tilde{\rho} - \rho$ with $\tilde{\rho} \in L_c^\infty(\mathcal{X}) \cap \mathcal{P}(\mathcal{X})$. We first compute

$$
\begin{aligned}
&\frac{d}{d\epsilon}\mathcal{F}(\rho + \epsilon\chi)\Big|_{\epsilon=0} \\
&= \frac{d}{d\epsilon}\left( \int R(x;\rho+\epsilon\chi)d\rho(x) + \epsilon\int R(x;\rho+\epsilon\chi)d\chi(x) \right. \\
&\qquad \left. - \int \log\pi(x)d\rho(x) - \epsilon\int \log\pi(x)d\chi(x) \right)\Big|_{\epsilon=0} \\
&= \left( \frac{d}{d\epsilon}\int R(x;\rho+\epsilon\chi)d\rho(x) + \int R(x;\rho+\epsilon\chi)d\chi(x) + \epsilon\frac{d}{d\epsilon}\int R(x;\rho+\epsilon\chi)d\chi(x) \right. \\
&\qquad \left. - \int \log\pi(x)d\chi(x) \right)\Big|_{\epsilon=0} \\
&= \int R(x;\rho) - \log\pi(x)d\chi(x) \\
&\qquad + \left( \int \frac{\partial}{\partial\epsilon}R(x;\rho+\epsilon\chi)d\rho(x) + \epsilon\int \frac{\partial}{\partial\epsilon}R(x;\rho+\epsilon\chi)d\chi(x) \right)\Big|_{\epsilon=0} \\
&= \int R(x;\rho) - \log\pi(x)d\chi(x) + \int \frac{\partial}{\partial\epsilon}R(x;\rho+\epsilon\chi)d(\rho+\epsilon\chi)(x)\Big|_{\epsilon=0}. \qquad (29)
\end{aligned}
$$

By assumption, the remainder term above is equal to zero and the functional derivative of $\mathcal{F}$ is therefore

$$
\frac{\delta\mathcal{F}}{\delta\rho}(\rho) = R_1(x;\rho) - \log\pi(x),
$$

so its Wasserstein gradient is

$$
\begin{aligned}
\nabla_W\mathcal{F}(\rho) &= \nabla\frac{\delta\mathcal{F}}{\delta\rho}(\rho) \\
&= \nabla R_1(x;\rho) - \nabla\log\pi(x).
\end{aligned}
$$

$\square$

*Proof of Proposition 3.9.* To show that $\rho^*$ is not the target, assume for the sake of a contradiction that $\rho^* = \pi$. So $\nabla_W \mathcal{F}(\pi) = 0$, and Proposition 3.8 gives

$$\frac{r(x; \pi)}{\pi(x)} = \nabla \log \pi(x)$$

$$r(x; \pi) = \nabla \pi(x)$$

$$\frac{\mathscr{F}(K_2)}{\mathscr{F}(K_1)} \mathscr{F}(\nabla \pi) = \mathscr{F}(\nabla \pi)$$

$$\mathscr{F}(K_2) = \mathscr{F}(K_1)$$

$$K_2 = K_1.$$

Therefore, it cannot be the case that $\rho^* = \pi$. $\qquad \square$

*Proof of Example 3.10.* A direct computation of (20) along with the definitions of $r$ and $R$ yields

$$\frac{\mathscr{F}(K_2)(\omega)}{\mathscr{F}(K_1)(\omega)} \cdot \mathscr{F}(\nabla \rho^*)(\omega) = \sqrt{\frac{h_2}{h_1}} \cdot \exp(-2\pi^2 \omega^2 \Delta h) \cdot 2\pi i \omega \exp(-2\pi^2 \omega^2 \sigma^2)$$

$$= \sqrt{\frac{h_2}{h_1}} \cdot 2\pi i \omega \cdot \exp(-2\pi^2 \omega^2 (\sigma^2 + \Delta h))$$

$$r(x; \rho^*) = -\sqrt{\frac{h_2}{h_1}} \cdot \frac{1}{\sqrt{2\pi}} \cdot \frac{x}{(\sigma^2 + \Delta h)^{3/2}} \cdot \exp\left(-\frac{x^2}{2(\sigma^2 + \Delta h)}\right)$$

$$\nabla R(x; \rho^*) = -\sqrt{\frac{h_2}{h_1}} \cdot \frac{\sigma}{(\sigma^2 + \Delta h)^{3/2}} \cdot x \exp\left(-\frac{x^2}{2(\sigma^2 + \Delta h)}\right) \cdot \exp\left(\frac{x^2}{2\sigma^2}\right)$$

$$= -\sqrt{\frac{h_2}{h_1}} \cdot \frac{\sigma}{(\sigma^2 + \Delta h)^{3/2}} \cdot x \exp\left(-\frac{\Delta h x^2}{2\sigma^2(\sigma^2 + \Delta h)}\right).$$

Now computing $\nabla \log \pi$, using some $A \in \mathbb{R}$ for the normalising constant, we have

$$\pi(x) = A \exp\left(-\alpha \exp\left(\frac{x^2}{2\beta}\right)\right)$$

$$\log \pi(x) = \log(A) - \alpha \exp\left(\frac{x^2}{2\beta}\right)$$

$$\nabla \log \pi(x) = -\frac{\alpha x}{\beta} \exp\left(\frac{x^2}{2\beta}\right)$$

$$= -\sqrt{\frac{h_2}{h_1}} \cdot \frac{\sigma}{(\sigma^2 + \Delta h)^{3/2}} \cdot x \exp\left(-\frac{\Delta h x^2}{2\sigma^2(\sigma^2 + \Delta h)}\right).$$

Since $\nabla R(x; \rho^*) = \nabla \log \pi(x)$, equation (21) implies that $\rho^*$ is a fixed point. $\qquad \square$

## B  Additional Experimental Results and Details

### B.1  Multivariate Gaussian Mixture

Figure 2 in Section 4 shows the projection onto the first two dimensions of particles sampled from high-dimensional Gaussian mixtures using different SVGD variants. The marginal density of the corresponding projection of the target density is overlaid. We extend this here with Figure 4, showing 5 additional projection plots of randomly selected pairs of dimensions between 1 and 1000.

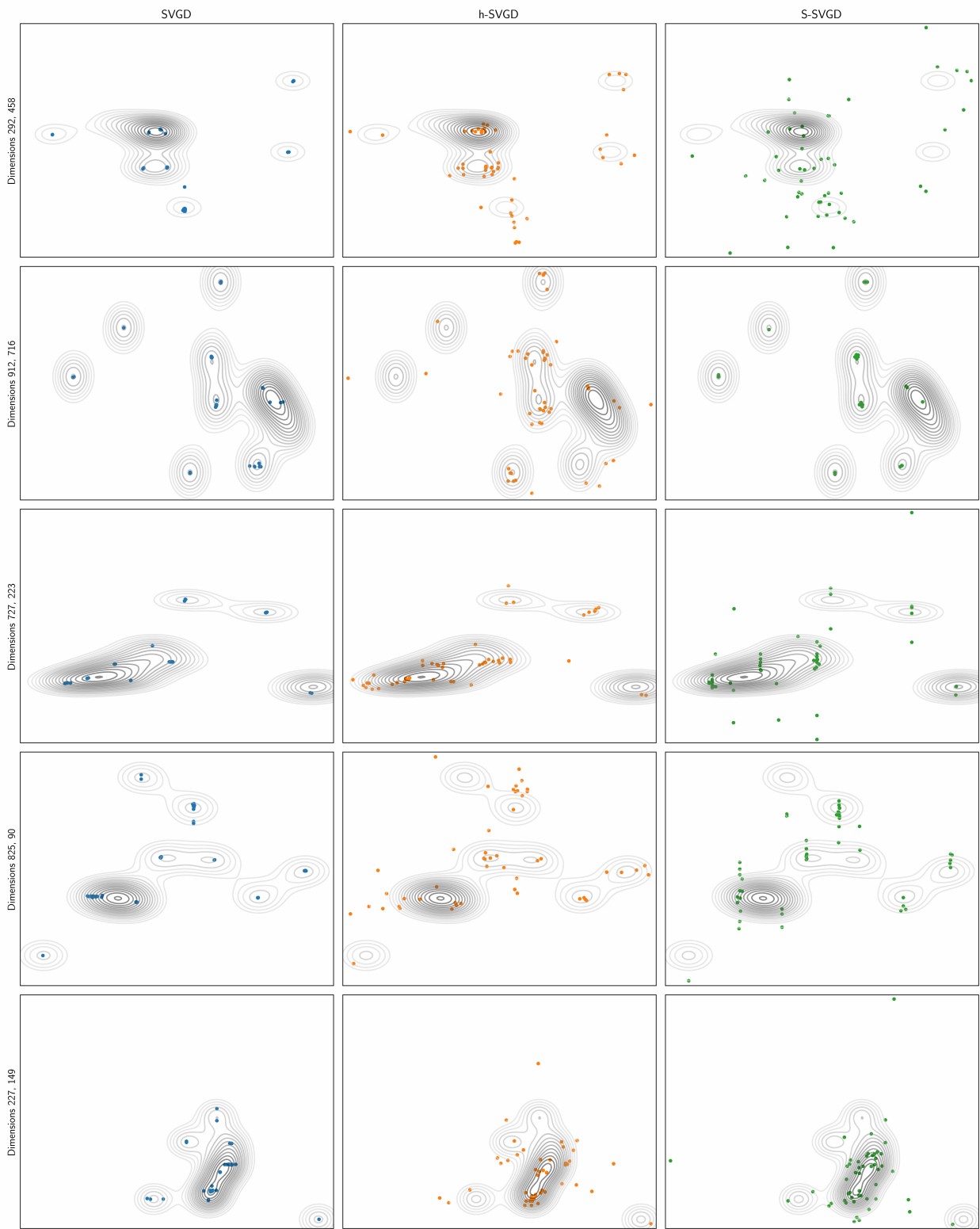

Figure 4: Projections onto 5 pairs of randomly selected dimensions of particles sampled from a high-dimensional Gaussian mixture using different SVGD variants. The contour plots of the corresponding marginal density are overlaid.

### B.2 Bayesian Neural Network

The results presented in Section 4.2 follow the settings of (Liu & Wang, 2016). In particular, we use Gaussian priors for the network weights and Gamma priors for the inverse covariances. There is one hidden layer with 50 units for most datasets, Protein and Year being the exceptions with 100 units each. The datasets are randomly partitioned into 90% for training and 10% for testing with results averaged over 20 trials, Protein and Year being the exceptions with 5 trials and 3 trials respectively. The number of particles in each case is 20, the activation function is $\text{RELU}(x) = \max(0, x)$, the number of iterations is 2000, and the mini-batch size is 100 for all datasets except for Year, which uses a mini-batch size of 1000.

We recreate Tables 1 and 2 below, this time comparing h-SVGD against SSVGD (Gong et al., 2021). Table 3 shows that h-SVGD outperforms SSVGD in mitigation of variance collapse on all but one dataset. We note that h-SVGD achieves this with a significantly faster runtime for all datasets, which is due to SSVGD requiring additional optimistaion of the projection matrix. Table 4 shows that SSVGD and h-SVGD are comparable in both RMSE and LL metrics. The h-SVGD algorithm achieves a better LL score on more datasets, but SSVGD achieves a better RMSE score on more datasets.

Table 3: DAMV and runtime in seconds of SSVGD and h-SVGD.

| Dataset | DAMV | | Runtime (seconds) | |
|---|---|---|---|---|
| | **SSVGD** | **h-SVGD** | **SSVGD** | **h-SVGD** |
| Boston | $0.035 \pm 0.002$ | $\mathbf{0.087 \pm 0.010}$ | $208 \pm 13$ | $\mathbf{28.5 \pm 0.9}$ |
| Concrete | $0.070 \pm 0.004$ | $\mathbf{0.102 \pm 0.006}$ | $148 \pm 56$ | $\mathbf{28.7 \pm 1.3}$ |
| Energy | $0.053 \pm 0.005$ | $\mathbf{0.106 \pm 0.011}$ | $156 \pm 28$ | $\mathbf{30.7 \pm 2.2}$ |
| Kin8nm | $0.083 \pm 0.002$ | $\mathbf{0.093 \pm 0.003}$ | $141 \pm 3.9$ | $\mathbf{36.0 \pm 1.3}$ |
| Naval | $\mathbf{0.070 \pm 0.021}$ | $0.068 \pm 0.011$ | $237 \pm 11$ | $\mathbf{34.5 \pm 0.9}$ |
| Combined | $0.118 \pm 0.005$ | $\mathbf{0.138 \pm 0.006}$ | $116 \pm 18$ | $\mathbf{36.7 \pm 2.4}$ |
| Protein | $0.057 \pm 0.006$ | $\mathbf{0.084 \pm 0.001}$ | $390 \pm 20$ | $\mathbf{72.8 \pm 1.7}$ |
| Wine | $0.029 \pm 0.002$ | $\mathbf{0.075 \pm 0.005}$ | $210 \pm 10$ | $\mathbf{29.8 \pm 1.1}$ |
| Yacht | $0.066 \pm 0.009$ | $\mathbf{0.121 \pm 0.012}$ | $97.9 \pm 1.2$ | $\mathbf{30.0 \pm 1.3}$ |
| Year | $0.012 \pm \text{NA}$ | $0.012 \pm \text{NA}$ | $12488 \pm \text{NA}$ | $\mathbf{666 \pm NA}$ |

Table 4: Average RMSE and LL of SSGVD and h-SVGD evaluated on the test dataset.

| Dataset | Test RMSE | | Test LL | |
|---|---|---|---|---|
| | **SSVGD** | **h-SVGD** | **SSVGD** | **h-SVGD** |
| Boston | $3.024 \pm 0.604$ | $\mathbf{3.001 \pm 0.584}$ | $-2.088 \pm 0.322$ | $\mathbf{-1.988 \pm 0.221}$ |
| Concrete | $\mathbf{5.073 \pm 0.522}$ | $5.210 \pm 0.529$ | $-2.563 \pm 0.239$ | $\mathbf{-2.535 \pm 0.179}$ |
| Energy | $0.923 \pm 0.123$ | $1.040 \pm 0.128$ | $\mathbf{-0.631 \pm 0.162}$ | $-0.805 \pm 0.104$ |
| Kin8nm | $\mathbf{0.084 \pm 0.003}$ | $0.090 \pm 0.003$ | $0.232 \pm 0.135$ | $\mathbf{0.468 \pm 0.090}$ |
| Naval | $\mathbf{0.003 \pm 0.000}$ | $0.004 \pm 0.000$ | $-0.624 \pm 0.161$ | $\mathbf{-0.090 \pm 0.105}$ |
| Combined | $\mathbf{4.028 \pm 0.220}$ | $4.057 \pm 0.218$ | $\mathbf{-2.335 \pm 0.066}$ | $-2.354 \pm 0.052$ |
| Protein | $\mathbf{4.581 \pm 0.026}$ | $4.600 \pm 0.026$ | $-2.526 \pm 0.045$ | $\mathbf{-2.456 \pm 0.017}$ |
| Wine | $0.676 \pm 0.051$ | $0.626 \pm 0.045$ | $-1.261 \pm 0.172$ | $\mathbf{-0.750 \pm 0.097}$ |
| Yacht | $\mathbf{1.664 \pm 0.607}$ | $1.861 \pm 0.662$ | $\mathbf{-0.788 \pm 0.511}$ | $-0.813 \pm 0.227$ |
| Year | $8.922 \pm \text{NA}$ | $\mathbf{8.689 \pm NA}$ | $-2.940 \pm \text{NA}$ | $\mathbf{-2.872 \pm NA}$ |

