# OpenReview forum: "Convergence Aspects of Hybrid Kernel SVGD"
_TMLR — Accepted by TMLR_

### Review · Reviewer_5Dn5 · 2025-08-31

**Summary Of Contributions:**

- This paper studies the theoretical, notably convergence, properties of Hybrid Kernel SVGD (h-SVGD) that was proposed by D’Angelo et al. (2021), where the attraction and repulsion kernels are two distinct kernels. In this setting, the infamous variance collapse issue can be mitigated and has empirically shown to be superior for high dimensional scenarios such as image classification
- This paper establishes the hybrid Stein operator and subsequently the hybrid Stein PDE, analogous to the Stein PDE of SVGD, under the mean field limit
- Regularity conditions are then shown and an equivalence with a Wasserstein gradient flow are shown when $k_1 = c * k_2$ for constant $c$
- The corresponding limiting distribution of the flow is given and it is proven that this is not equal to the target distribution $\pi$
- Furthermore, dissipation results are also shown for the proposed Wasserstein gradient flow function
- A more general theoretical result is derived for any $k_1$ and $k_2$, and the theory shows the difficulties in obtaining feasible kernels.
- A toy example is given that shows this inequality.
- Finally, numerical results show that despite not having a mean field limiting distribution of the target distribution, h-SVGD surpasses the performance of SVGD. The theory suggests a scaling factor of $c=d^\alpha$ for some $\alpha\in (0,1)$ to help alleviate variance collapse. Moreover, using DAMV as a measure of spread reviews that h-SVGD particles are more spread out than SVGD and SSVGD

**Audience:**

Yes

**Audience Explanation:**

- Variance collapse issue is one of the main obstacles for SVGD that stops it from being widely adopted by the Bayesian machine learning community, especially for high-dimensional tasks such as BNN inference.
- This paper shows an important theoretically-backed result for h-SVGD, one of the popular methods out there that has shown promise in the high-dimensional setting
- Past and recent works such as Sliced SVGD, Grassmann SVGD, Projected SVGD etc… have all attempted to tackle the issue of variance collapse as well, so there is ample interest and room for improvement in this topic.

**Broader Impact Concerns:**

This is mostly theoretical work, no concerns that I am aware of.

**Claims And Evidence:**

Yes

**Claims Explanation:**

- The theoretical claims are accurate upon checking the proofs. Some of the proofs rely on the theorems and proofs in Lu et al. (2019) and Korba et al. (2020), where the setting is the classic SVGD setting with a single kernel $k$. The authors modify the proof techniques with $k_1$ and $k_2$ and obtain similarly-looking results.
- Example 3.10 is a nice illustration of what the limiting distribution $\rho^*$ is and how it is not equal to the target distribution
- Although DAMV has been used in various works, such as in Sliced SVGD and Message Passing SVGD, it simply measures the spread of the particles, and not whether the inferred distribution matches (or is spread out according to) the target distribution. I would suggest also looking at the energy distance as utilized in Grassmann SVGD (Liu et al. 2022)., or looking at the inferred covariance matrix (for the synthetic experiment) and using HMC samples as a reference ground-truth distribution.

**Requested Changes:**

I would suggest considering extra measures of “goodness-of-spread” as I suggested above, such as the energy distance, inferred covariance matrix or using HMC samples.

---

> ### Author Response · Authors · 2025-11-12
> **Main Response**
>
> We thank the reviewer for their feedback and suggestions on the experiments. We have uploaded a revised manuscript with changes in red.
>
> ## Requested Changes:
> - We have now added an extra chart to Figure 1 that shows the energy distance between samples and the target distribution for the multivariate Gaussian mixture experiment in Section 4.1. We have also updated the discussion to reflect the results that h-SVGD outperforms both SVGD and SSVGD in terms of energy distance.
> - We have also computed energy distances for the Bayesian neural network experiment in Section 4.2. Our ground truth for the energy distance calculation is 10,000 samples generated by the no-U-turn sampler (NUTS) [1], which has been used in other SVGD applications such as Bayesian logistic regression [2] and graph applications [3]. We have added the energy distance comparison between SVGD variants in Figure 3 and found that h-SVGD beats SVGD on all but one dataset, and beats SSVGD on all but two datasets.
>
>
> ## References:
> - [1] Hoffman, M.D. and Gelman, A., 2014. The No-U-Turn sampler: adaptively setting path lengths in Hamiltonian Monte Carlo. J. Mach. Learn. Res., 15(1), pp.1593-1623.
> - [2] Liu, X., Zhu, H., Ton, J.F., Wynne, G. and Duncan, A., 2022. Grassmann Stein variational gradient descent. arXiv preprint arXiv:2202.03297.
> - [3] Wang, D., Zeng, Z. and Liu, Q., 2018, July. Stein variational message passing for continuous graphical models. In International Conference on Machine Learning (pp. 5219-5227). PMLR.

---

> > ### Comment · Reviewer_5Dn5 · 2025-11-23
> > **Reply**
> >
> > Thank you so much for addressing all my comments. I am happy with the revised manuscript.

---

### Review · Reviewer_iHeY · 2025-10-28

**Summary Of Contributions:**

Stein variational gradient descent (SVGD) is a non-parametric variational inference framework, which has gathered significant attention for appealing empirical and theoretical performance. A variant of SVGD, called hybrid kernel Stein variational gradient descent (h-SVGD), uses two different kernels for "driving" and "repulsive" terms of SVGD respectively. The variant, h-SVGD, can mediate a well-known issue of SVGD in underestimating the variance of a target distribution, by controlling the balance between the driving and repulsive terms. The focus of the paper is on theoretical analysis of h-SVGD, which had not been studied before. In particular, the paper shows that an updating scheme, or a flow, of densities defined by h-SVGD corresponds to minimisation of some functional over the space of densities, similarly to original SVGD. However, due to the use of two different kernels, the density to which this flow converges is no longer a target density, unlike SVGD. The numerical analysis provided in the paper illustrates that h-SVGD estimates the variance of a target distribution better than original SVGD.

The main part of the theoretical analysis is to reproduce the result of [1] and [2], in the setting that the kernels of the driving and repulsive terms are different from each other, as well as to study the limiting density of the flow. The strength is in confirming that h-SVGD indeed cannot converge to a given target density due to the use of two kernels, while it still defines a flow that minimises some functional. The weakness may be in that some main results themselves could be a relatively straightforward extension of [1] and [2], as we can apply their analysis to each driving and repulsive term separately, with a different kernel used.

[1] Anna Korba, Adil Salim, Michael Arbel, Giulia Luise, and Arthur Gretton. A non-asymptotic analysis for
Stein variational gradient descent. Advances in Neural Information Processing Systems, 33:4672–4682,
2020.

[2] Jianfeng Lu, Yulong Lu, and James Nolen. Scaling limit of the Stein variational gradient descent: The mean
field regime. SIAM Journal on Mathematical Analysis, 51(2):648–671, 2019.

**Audience:**

Yes

**Audience Explanation:**

The idea of h-SVGD to use two different kernels for the driving and repulsive terms in SVGD is an interesting idea to control the balance between the two terms. Understanding what properties are same as, or different from, original SVGD would be an interesting question for the community working on SVGD.

**Broader Impact Concerns:**

No concern on the ethical implication.

**Claims And Evidence:**

Yes

**Claims Explanation:**

The proof seems correct to me. Proposition 3.1, 3.3, and 3.7 essentially follow from [1] and [2]. The rest follows from standard calculus, e.g., to derive Wasserstein gradients and time derivatives.

**Requested Changes:**

### Major Point

- The terminology, driving and repulsive terms, are fine by readers who are familiar with the context. Though, for those who are not familiar, it may be worth concisely explaining what they are. For example, something like (2) in [3] would suffice.
- It would be worth containing a definition and concise explanation about the first variation (which is referred to as functional derivative) and the Wasserstein gradient in P.6.
- It would be nice to define the Fourier transform and the inverse somewhere either in Appendix or main text.
- Readers may benefit some 2D illustrative figure to see how h-SVGD samples differ from standard SVGD samples. Currently, only the graphs and numbers of the error/variance estimate are provided.
- It seems to me that one finding in this paper can suggest that the use of tempered density is helpful to mediate the variance-underestimation of SVGD, which makes sense. If this is correct, it may be worth adding some discussion in the paper.
- In Corollary 3.5, the constant c should be $c \ne 1$?
- In Proposition 3.8, since the condition of the kernels are mentioned in the main text, it may be nice to clarify the condition under which the inverse Fourier transform exists in (20) rather than directly assuming the existence.



### Minor Point

- Abstract: particle based -> particle-based?
- P.1: perhaps \cite should be used for Duncan et al.
- P.1: perhaps \cite should be used for Zhou et al.
- P.7: perhaps \Cref should be used for Proposition (3.4).
- P.9: such a a -> such a



[3] Jimmy Ba, Murat A Erdogdu, Marzyeh Ghassemi, Shengyang Sun, Taiji Suzuki, Denny Wu, and Tianzong
Zhang. Understanding the variance collapse of SVGD in high dimensions. In International Conference on
Learning Representations, 2021.

---

> ### Author Response · Authors · 2025-11-12
> **Main Response**
>
> We thank the reviewer for their feedback and detailed suggestions. We have uploaded a revised manuscript with changes in red.
>
> ## Major Points:
> - We have now added underbraces to clearly indicate which terms in Equations (4) and (5) correspond to the driving and repulsive forces. Along with the definitions of $\bf G$ and $\bf R$ at the beginning of Section 3, this should make it more clear for those not yet familiar with the terminology.
> - We have now added a definition of the functional derivative along with a reference at the begging of Section 3.4.
> - We have now added the definition of the Fourier transform and its inverse to the proof of Proposition 3.8 in Appendix A. We make this reference to the appendix just after Equation (20), where the Fourier transform is first used in the text.
> - We have now added some 2D scatter plots for the BNN experiment, both in Section 4.1 and Appendix B.
> - We have now added a note in the conclusion about tempered distributions as a direction for future work.
> - In Corollary 3.5, our new result is indeed for $c\neq1$. This extends the $c=1$ case (Liu and Wang, 2016). After consideration we chose to keep $c>0$ in the statement of Corollary 3.5 to emphasise that SVGD converges to the target (i.e. $\rho^*=\pi$) only when $c=1$. We have added a note in the paragraph after the statement of the corollary to hightlight that generalisation.
> - We have now added some discussion about conditions for existence of the inverse Fourier transform after Example 3.10 as it is pertinent to that kernel choice.
>
> ## Minor Points:
> - We have updated "particle based" to "particle-based" in the abstract.
> - We have updated the references to Duncan et al. and Zhou et al. in the Introduction to use the `\citet` command.
> - We have removed the parentheses from the reference to Poposition 3.4 in the first sentence of Section 3.5.
> - We have removed the sentence with the "such a a" typo that was previously on page 9 because we have added a comment on this direction for future work to the beginning of Section 4.

---

> > ### Comment · Reviewer_iHeY · 2025-11-13
> > **Reviewer Response**
> >
> > Thank you so much for addressing all my comments. I am personally happy with the updated manuscript.

---

### Review · Reviewer_XMDu · 2025-10-29

**Summary Of Contributions:**

This paper provides a rigorous theoretical and empirical analysis of hybrid kernel SVGD (h-SVGD). The authors frame its mean-field limit as a kernelized Wasserstein gradient flow on a functional other than the Kullback-Leibler divergence. Theoretically, they demonstrate that h-SVGD does not converge to the target distribution $\pi$ in this limit. Despite this negative asymptotic result, the paper's key practical contribution is its demonstration that h-SVGD effectively mitigates the variance collapse that plagues standard SVGD in high dimensions. Crucially, this improved variance estimation is achieved at a negligible additional computational cost, making h-SVGD a computationally efficient alternative to other methods that also address variance collapse.

**Audience:**

Yes

**Audience Explanation:**

Methods based on SVGD are widely used, and this paper provides a valuable analysis of the h-SVGD variant. The TMLR audience will be interested in the paper's central, and somewhat paradoxical, finding: while the authors prove theoretically that h-SVGD does not converge to the target distribution in the mean-field limit , they also demonstrate empirically that it effectively solves the practical problem of variance collapse in high dimensions. The finding that it provides this benefit at almost no additional computational cost  makes this a result of clear practical interest.

**Broader Impact Concerns:**

This paper is a foundational, theoretical work on algorithm properties, not a specific real-world application. As such, it does not present any direct broader impact concerns. Any potential societal impacts would be indirect and depend on future downstream applications, which are outside the scope of this submission.

**Claims And Evidence:**

Yes

**Claims Explanation:**

The paper successfully uses mathematical proof to establish its theoretical (asymptotic) results. It then uses clear, targeted experiments to demonstrate the algorithm's practical utility in the finite-particle, high-dimensional regime—a setting where the asymptotic assumptions are no longer met and finite-particle effects dominate. The evidence for both of these complementary aspects is accurate and clearly presented.

**Requested Changes:**

1. The paper's main conclusion rests on a disconnect between the mean-field theory and the practical results. The gradient flow framework suggests convergence to a "smoothed" distribution $\rho^{*}(x) \propto \pi(x)^{1/c}$ (Corollary 3.5) when $k_2=c k_1$, but the Gaussian mixture experiments show the particle variance converging to the true target variance, not a scaled one. The paper would be far more convincing if the authors could provide a more principled justification, even if not a full proof, for why this heuristic scaling in the finite-particle regime so neatly "corrects" the variance, thereby bridging the gap between the mean-field theory and the practical experiments.

2. In (B2), "such that both $k\_1$ and $\nabla k\_2$ and are $D$-Lipschitz" contains a typo.

3. The derivation from (10) to (11) appears to be missing a step, such as an integration by parts, that would be necessary to move from $\nabla_{x_j}k_2$ to $T_{k_2,\rho}(\nabla \log \rho)$. I suggest the authors complete the computation for better completeness. And the definition of $T_{k,\rho}$ is not consistently written.

4. The repulsive force $R(\cdot;k,\mu)$ is not defined before being used in the first paragraph of Section 3.4; please define it previously.

---

> ### Author Response · Authors · 2025-11-12
> **Main Response**
>
> We thank the reviewer for their feedback and detailed suggestions. We have uploaded a revised manuscript with changes in red.
>
> ## Requested Changes:
> - We have added in some additional discussion on the motivation and justification for the kernel scaling heuristic at the beginning of Section 4. We use two facts: The repulsive force of SVGD under an RBF kernel decreases as $||{\bf R}(\ \cdot\ ; k, \mu)||_\infty = O(d^{-\alpha})$ [1], and scaling the repulsive kernel as $k_2=ck_1$ for a normal target density with $c>0$ will scale the variance of the mean field limit by $c$ (Corollary 3.5). These two facts together justify the choice of $k_2=d^\alpha k_1$ for the repulsive kernel in being able to offset the variance underestimation in high dimensions with finite particles.
> - We have removed the additional "and" from Assumption (B2).
> - We have added an extra line in between Equations (10) and (11) to make the derivation more clear. In particular, we applied integration by parts and noted the use of the identity $\nabla\rho = \rho\nabla\log\rho$ in arriving at Equation (11). We have also updated the definition of $T_{k,\rho}$ to make the use of $x$ and $x'$ consistent with Equations (10) and (11).
> - We have added a comment at the beginning of Section 3.4 referencing the definition of $\bf R$ made earlier in Section 3.1, just after Equation (6). This should assist readers in recalling the definition for this discussion.
>
> ## References:
> [1] Jingwei Zhuo, Chang Liu, Jiaxin Shi, Jun Zhu, Ning Chen, and Bo Zhang. Message passing Stein variational
> gradient descent. In International Conference on Machine Learning, pp. 6018–6027. PMLR, 2018.

---

### Decision · Action_Editor_WLBA · 2025-11-27

**Recommendation:** Accept as is

**Audience:**

Yes

**Audience Explanation:**

People working on Bayesian inference and sampling methods would be interested in this paper. SVGD variance collapse is a known issue and this paper provides promising results towards addressing it.

**Claims And Evidence:**

Yes

**Claims Explanation:**

The paper proposed a new variant of SVGD called h-SVGD, where two different kernels are used for the driving and repulsive terms. The theoretical analysis in this work is of interest to communities working on SVGD, showing that h-SVGD is a promising method that can alleviate the variance collapse issue of SVGD in high dimensions. Numerical results on high-dim mixture of Gaussians and Bayesian neural networks are provided to demonstrate the claims.

Most of reviewers' comments and questions have been addressed in the author feedback.